# Implicit regularization of multi-task learning and finetuning in overparameterized neural networks

## Abstract

It is common in deep learning to train networks on auxiliary tasks with the expectation that the learning will transfer, at least partially, to another task of interest. In this work, we investigate the inductive biases that result from learning auxiliary tasks, either simultaneously (multi-task learning, MTL) or sequentially (pretraining and subsequent finetuning, PT+FT). In the simplified setting of two-layer diagonal linear networks trained with gradient descent, we apply prior theoretical results to describe implicit regularization penalties associated with MTL and PT+FT, both of which incentivize feature sharing between tasks and sparsity in learned task-specific features. Notably, these results imply that during finetuning, networks operate in a hybrid of the kernel (or "lazy") regime and the feature learning ("rich") regime identified in prior work. Moreover, we show that PT+FT can exhibit a novel "nested feature selection" behavior not captured by either regime, which biases it to extract a sparse subset of the features learned during pretraining. In ReLU networks, we reproduce all of these qualitative behaviors empirically, in particular verifying that analogues of the sparsity biases predicted by the linear theory hold in the nonlinear case. We also observe that PT+FT (but not MTL) is biased to learn features that are correlated with (but distinct from) those needed for the auxiliary task, while MTL is biased toward using identical features for both tasks. As a result, we find that in realistic settings, MTL generalizes better when comparatively little data is available for the task of interest, while PT+FT outperforms it with more data available. We show that our findings hold qualitatively for a deep architecture trained on image classification tasks. Our characterization of the nested feature selection regime also motivates a modification to PT+FT that we find empirically improves performance. Overall, our results shed light on the impact of auxiliary task learning and suggest ways to leverage it more effectively.

## 1 Introduction

Neural networks are often trained on multiple tasks, either simultaneously ("multi-task learning," henceforth MTL, see Vafaeikia et al. (2020); Zhang & Yang (2022)) or sequentially ("pretraining" and subsequent "finetuning," henceforth PT+FT, see Du et al. (2022); Zhou et al. (2023)). Empirically, models are able to transfer knowledge from auxiliary tasks to improve performance on tasks of interest. However, understanding of how auxiliary tasks influence learning remains limited.

Auxiliary tasks are especially useful when there is less data available for the target task. Modern "foundation models," trained on data-rich general-purpose auxiliary tasks (like next-word prediction or image generation) before adaptation to downstream tasks, are a timely example of this use case (Bommasani et al., 2022). Auxiliary tasks are also commonly used in reinforcement learning, where performance feedback can be scarce (Jaderberg et al., 2016). Intuitively, auxiliary task learning biases the target task solution to use representations shaped by the auxiliary task. When the tasks share common structure, this influence may enable generalization from relatively few training samples for the task of interest. However, it can also have downsides, causing a model to inherit undesirable biases from auxiliary task learning (Wang & Russakovsky, 2023; Steed et al., 2022).

A relevant insight from the literature on single-task learning is that a combination of initialization and learning dynamics produces an implicit regularizing effect on learned solutions. This regularization can enable good generalization even when models are overparameterized (Neyshabur, 2017).

**Contributions.** In this work we characterize the inductive biases of MTL and PT+FT in terms of implicit regularization. Note that we focus on MTL in which feature extraction layers are shared and readouts are task-specific, and on PT+FT in which the readout of the network is reinitialized before finetuning. We first apply prior theoretical results that study a simplified "diagonal linear network" model (which importantly still captures a notion of feature learning/selection) to the settings of PT+FT and MTL. These results provide an exact description of the solutions learned by PT+FT in diagonal linear networks, and an approximate description of those learned via MTL, in terms of norm minimization biases. Both biases encourage (1) the reuse of auxiliary task features and (2) sparsity in learned task-specific features. For PT+FT, this bias corresponds to a hybrid of "rich" and "lazy" learning dynamics in different parts of the network. Additionally, we find that under suitable parameter scalings, PT+FT exhibits a novel "nested feature-selection" regime, distinct from previously characterized rich and lazy regimes, which biases finetuning to extract sparse subsets of the features learned during pretraining. In ReLU networks, we reproduce these phenomena empirically. Based on the nested feature selection insight, we suggest practical tricks to improve finetuning performance, which shows positive results in experiments. We also describe a qualitative behavior of PT+FT not captured by the linear theory: a bias toward learning main task features correlated with (but not necessarily identical to) those learned during pretraining, which we find is beneficial given sufficient training data for the task of interest but can be detrimental when data is scarce.

## 2 RELATED WORK

A variety of studies have characterized implicit regularization effects in deep learning. These include biases toward low-frequency functions (Rahaman et al., 2018), toward stable minima in the loss landscape (Mulayoff et al., 2021), toward low-rank solutions (Huh et al., 2023), and toward lower-order moments of the data distribution (Refinetti et al., 2023). In shallow (single hidden-layer) networks, Chizat & Bach (2020) show that when using cross-entropy loss, shallow networks are biased to minimize the $\mathcal{F}_1$ norm, an infinite-dimensional analogue of the $\ell_1$ norm over the space of possible hidden-layer features (see also Lyu & Li, 2020; Savarese et al., 2019). Other work has shown that implicit regularization for mean squared error loss in nonlinear networks cannot be exactly characterized as norm minimization (Razin & Cohen, 2020), though $\mathcal{F}_1$ norm minimization is a precise description under certain assumptions on the inputs (Boursier et al., 2022).

Compared to the body of work on inductive biases of single-task learning, theoretical treatments of MTL and PT+FT are more scarce. Some prior studies have characterized benefits of multi-task learning with a shared representational layer in terms of bounds on sample efficiency (Maurer et al., 2016; Wu et al., 2020). Others have characterized the learning dynamics of linear networks trained from nonrandom initializations, which can be applied to understand finetuning dynamics (Braun et al., 2022; Shachaf et al., 2021). However, while these works demonstrate an effect of pretrained initializations on learned solutions, the linear models they study do not capture the notion of feature learning we are interested in. A few empirical studies have compared the performance of multi-task learning vs. finetuning in language tasks, with mixed results depending on the task studied (Dery et al., 2021; Weller et al., 2022). Several authors have also observed that PT+FT outperforms PT + "linear probing" (training only the readout layer and keeping the previous layers frozen at their pretrained values), implying that finetuning benefits from the ability to learn task-specific features (Kumar et al., 2022; Kornblith et al., 2019b).

### 2.1 INDUCTIVE BIASES OF DIAGONAL LINEAR NETWORKS

The theoretical component of our study relies heavily on a line of work (Woodworth et al., 2020; Pesme et al., 2021; Azulay et al., 2021; HaoChen et al., 2021; Moroshko et al., 2020) that studies a inductive biases of a simplified "diagonal linear network" model. Diagonal linear networks parameterize linear maps $f : \mathbb{R}^d \to \mathbb{R}$ as

$$f_{\vec{w}}(\vec{x}) = \vec{\beta}(\vec{w}) \cdot \vec{x}, \qquad \beta_d(\vec{w}) := w^{(2)}_{+,d} w^{(1)}_{+,d} - w^{(2)}_{-,d} w^{(1)}_{-,d} \tag{1}$$

where $\vec{\beta}(\vec{w}) \in \mathbb{R}^D$. These correspond to two-layer linear networks in which the first layer consists of one-to-one connections, with duplicate $+$ and $-$ pathways to avoid saddle point dynamics around

$\vec{w} = 0$. Woodworth et al. (2020) showed that overparameterized diagonal linear networks trained with gradient descent on mean squared error loss find the zero-training-error solution that minimizes $\|f\|_{\ell_2}^2 = \sum_{d=1}^{D} \beta_d^2$, when trained from large initialization (the "lazy" regime, equivalent to ridge regression). When trained from small initialization, networks instead minimize $\|f\|_{\ell_1} = \sum_{d=1}^{D} |\beta_d|$ (the "rich" regime). The latter $\ell_1$ minimization bias is equivalent to minimizing the $\ell_2$ norm of the parameters $\vec{w}$ (Appendix B). This bias is a linear analogue of feature-learning/feature-selection, as a model with an $\ell_1$ penalty tends to learn solutions that depend on a sparse set of input dimensions.

## 3   THEORY OF PT+FT AND MTL IN DIAGONAL LINEAR NETWORKS

### 3.1   FINETUNING COMBINES RICH AND LAZY LEARNING

We now consider the behavior of PT+FT in overparameterized diagonal linear networks trained to minimize mean-squared error loss using gradient flow. We assume that all network weights are initialized prior to pre-training with a constant magnitude $\alpha$. We further assume that during pretraining, network weights are optimized to convergence on the training dataset $(X^{aux}, \vec{y}^{aux})$ from the auxiliary task, then the second-layer weights $(w_{+,d}^{(2)}$ and $w_{-,d}^{(2)})$ are reinitialized with constant magnitude $\gamma$, and the network weights are further optimized to convergence on the main task dataset $(X, \vec{y})$. The dynamics of the pretraining and finetuning steps can be derived as a corollary of the results of Woodworth et al. (2020) and Azulay et al. (2021):

**Corollary 1.** *If the gradient flow solution $\vec{\beta}^{aux}$ for the diagonal linear model in Eq. 1 during pretrainig fits the auxiliary task training data with zero error (i.e. $X^{aux}\vec{\beta}^{aux} = \vec{y}^{aux}$), and following reinitialization of the second-layer weights and finetuning, the gradient flow solution $\vec{\beta}^*$ after finetuning fits the main task data with zero training error (i.e. $X\vec{\beta} = \vec{y}$), then*

$$\vec{\beta}^* = \arg\min_{\vec{\beta}} \|\vec{\beta}\|_Q \quad s.t. \quad X\vec{\beta} = \vec{y}, \tag{2}$$

$$\|\vec{\beta}\|_Q := \sum_{d=1}^{D} \left(|\beta_d^{aux}| + \gamma^2\right) q\left(\frac{2\beta_d}{|\beta_d^{aux}| + \gamma^2}\right), \qquad q(z) = 2 - \sqrt{4 + z^2} + z \cdot arcsinh(z/2), \tag{3}$$

It is informative to consider limits of the expression 3. As $\frac{|\beta_d|}{|\beta_d^{aux}| + \gamma^2} \to \infty$, the contribution of a feature $d$ approaches $c|\beta_d|$ where $c \sim \mathcal{O}\left(\log\left(1/(|\beta_d^{aux}| + \gamma^2)\right)\right)$. As $\frac{|\beta_d|}{|\beta_d^{aux}| + \gamma^2} \to 0$, the contribution converges to $\beta_d^2/|\beta_d^{aux}|$. Thus, for features that are weighted sufficiently strongly by the auxiliary task (large $|\beta_d^{aux}|$), finetuning minimizes a weighted $\ell_2$ penalty that encourages reuse of features in proportion to their auxiliary task weight. For features specific to the auxiliary task (low $|\beta_d^{aux}|$), finetuning is biased to minimize an $\ell_1$ penalty, encouraging sparsity in task-specific features. Overall, the penalty decreases with $|\beta_d^{aux}|$, encouraging feature reuse where possible. The combination of $\ell_1$ and $\ell_2$ behavior, as well as the dependence on $|\beta_d^{aux}|$, can be observed in Fig. 1a (left panel).

### 3.2   MULTI-TASK TRAINING LEARNS SPARSE AND SHARED FEATURES

Now we consider MTL for diagonal linear networks. A multi-output diagonal linear network with $O$ outputs can be written as

$$f_{\vec{w}}(\vec{x}) = \beta(\vec{w})\vec{x}, \vec{\beta}_o(\vec{w}) := \vec{w}_{+,o}^{(2)} \circ \vec{w}_+^{(1)} - \vec{w}_{-,o}^{(2)} \circ \vec{w}_-^{(1)} \tag{4}$$

where $\beta(\vec{w}) \in \mathbb{R}^{O \times D}$, and $\circ$ is elementwise multiplication. We consider the effect of minimizing $\|\vec{w}\|_2$, as an approximation of the inductive bias of training a network from small initialization. We argue that $\|\vec{w}\|_2$ minimization is a reasonable heuristic. First, the analogous result holds in the single-output case for infinitesimally small initialization and two layers (though not for deeper networks, see Woodworth et al. (2020)). Second, for cross-entropy loss it has been shown that gradient flow converges to a KKT point of a max-margin/min-parameter-norm objective (Lyu & Li, 2020). Finally, explicit $\ell_2$ parameter norm regularization ("weight decay") is commonly used.

In MTL, a result of Dai et al. (2021) shows that a parameter norm minimization bias translates to minimizing an $\ell_{1,2}$ penalty that incentivizes group sparsity (Yuan & Lin, 2006) on the learned linear map $\beta$: $\|\beta\|_{1,2} := 2 \sum_{d=1}^{D} \|\vec{\beta}_{.,d}\|_2$ (a self-contained proof is given in Appendix B). For the specific

case of two outputs corresponding to main (first index) and auxiliary (second index) tasks, we have:

**Corollary 2.** *Using the multi-output diagonal linear model of Eq. 4 with two outputs, adopting shorthand notation $\vec{\beta} := \vec{\beta_1}$, $\vec{\beta}^{aux} := \vec{\beta_2}$, a solution $\beta^*$ with minimal parameter norm $||\vec{w}_+^{(1)}||_2^2 + ||\vec{w}_-^{(1)}||_2^2 + \sum_o ||\vec{w}_{+,o}^{(1)}||_2^2 + \sum_o ||\vec{w}_{-,o}^{(1)}||_2^2$ subject to the constraint that it fits the training data ($X\vec{\beta} = \vec{y}$, $X^{aux}\vec{\beta}^{aux} = \vec{y}^{aux}$) also minimizes the following:*

$$\beta^* = \arg\min_{\beta} \left( 2 \sum_{d=1}^D \sqrt{(\beta_d^{aux})^2 + (\beta_d)^2} \right) \quad s.t. \quad X\vec{\beta} = \vec{y}, \ X^{aux}\vec{\beta}^{aux} = \vec{y}^{aux}. \quad (5)$$

This penalty (plotted in Fig. 1a, right panel), encourages using shared features for the main and auxiliary tasks, as the contribution of $\beta_d$ to the square-root expression is smaller when $\beta_d^{aux}$ is large. As $\frac{|\beta_d|}{|\beta_d^{aux}|+\gamma^2} \to \infty$, the penalty converges to $2|\beta_d|$, a sparsity-inducing $\ell_1$ bias for task-specific features. As $\frac{|\beta_d|}{|\beta_d^{aux}|+\gamma^2} \to 0$ it converges to $\frac{\beta_d^2}{|\beta_d^{aux}|}$, a weighted $\ell_2$ bias as in the PT+FT case.

### 3.3 COMPARISON OF THE MTL AND PT+FT BIASES

We now compare the MTL and PT+FT penalties given above.[1] The MTL and PT+FT penalties have many similarities. Both decrease as $|\beta_d^{aux}|$ increases, both are proportional to $|\beta_d|$ as $\frac{|\beta_d|}{|\beta_d^{aux}|+\gamma^2} \to \infty$, and both are proportional to $\frac{\beta_d^2}{|\beta_d^{aux}|}$ as $\frac{|\beta_d|}{|\beta_d^{aux}|+\gamma^2} \to 0$. These similarities are evident in Fig. 1a. However, two differences between the penalties merit attention.

First, the relative weights of the $\ell_1$ and weighted $\ell_2$ penalties are different between MTL and PT+FT. In particular, in the $\ell_1$ penalty limit, there is an extra factor of order $\mathcal{O}\left(\log\left(1/(|\beta_d^{aux}| + \gamma^2)\right)\right)$ in the PT+FT penalty. Assuming small initializations, this factor tends to be larger than 2, the corresponding coefficient in the MTL penalty. Thus, PT+FT is more strongly biased toward reusing features from the auxiliary task (i.e. features where $\beta_d^{aux} \gg 0$) compared to MTL. We are careful to note, however, that in the case of nonlinear networks this effect is complicated by a qualitatively different phenomenon with effects in the reverse direction (see Section 5.2).

Second, the two norms behave differently for intermediate values of $\frac{\beta_d}{|\beta_d^{aux}|}$. In particular, as $\beta_d$ increases beyond the value of $\beta_d^{aux}$, the MTL norm quickly grows insensitive to the value of $\beta_d^{aux}$ (Fig. 1a, right panel). On the other hand, the PT+FT penalty remains sensitive to the value of $\beta_d^{aux}$ even for fairly large values of $\beta_d$, well into the $\ell_1$-like penalty regime (Fig. 1a, left panel). This property of the PT+FT norm, in theory, can enable finetuned networks to exhibit a rich regime-like sparsity bias while remaining influenced by their initializations. We explore this effect in section 4.2.

## 4 VERIFICATION AND IMPLICATIONS OF THE LINEAR THEORY

To validate these theoretical characterizations and illustrate their consequences, we performed experiments with diagonal linear networks in a teacher-student setup. We consider linear regression tasks defined by $\vec{w} \in \mathbb{R}^{1000}$ with a sparse set of $k$ non-zero entries. We sample two such vectors, corresponding to "auxiliary" and "main" tasks, varying the number of nonzero entries $k_{aux}$ and $k_{main}$, and the number of shared features (overlapping nonzero entries). We train diagonal linear networks on data generated from these ground-truth weights, using 1024 auxiliary task samples and varying the number of main task samples. For re-initialization of the readout, we use $\gamma = 10^{-3}$.

### 4.1 FEATURE REUSE AND SPARSE TASK-SPECIFIC FEATURE SELECTION IN PT+FT AND MTL

We begin with tasks in which $k_{aux} = k_{main} = 40$ (both tasks use the same number of features), varying the overlap between the feature sets (Fig. 1b). Both MTL and PT+FT display greater sample efficiency than single-task learning when the feature sets overlap. This behavior is consistent with an inductive bias towards feature sharing. Additionally, both MTL and PT+FT substantially outperform single-task lazy-regime learning, and nearly match single-task rich-regime learning, when the feature sets are disjoint. This is consistent with the $\ell_1$-like biases for task-specific features derived

---

[1]We note that it is not a strict apples-to-apples comparison as the PT+FT penalty describes the bias of gradient flow, while the given MTL penalty describes the bias of $\|\vec{w}\|_2$ minimization.

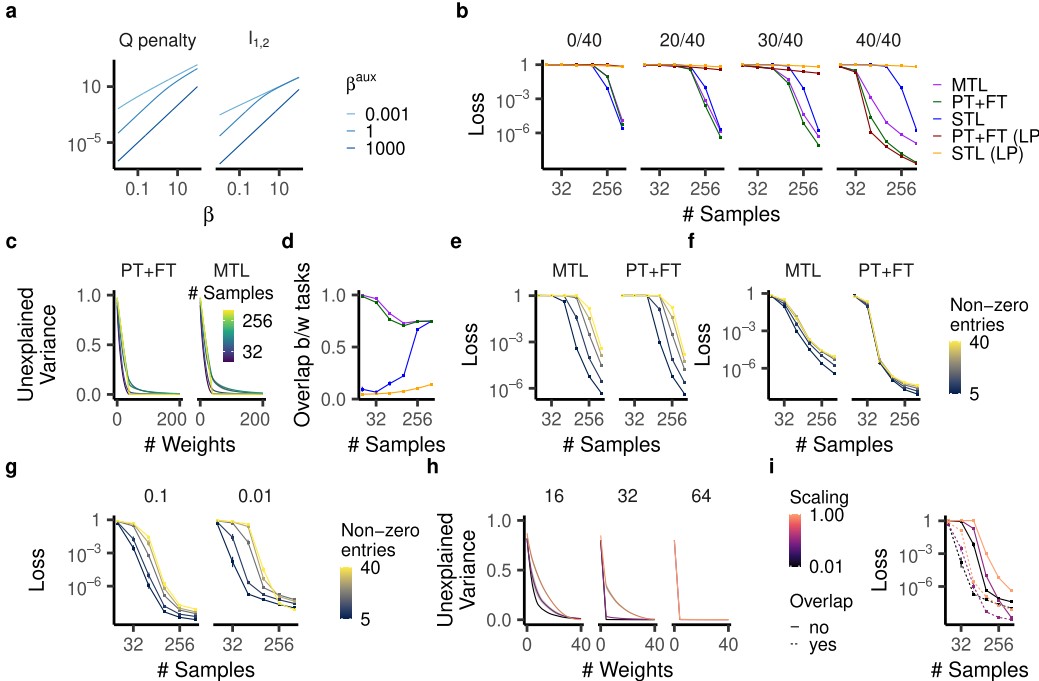

Figure 1: Diagonal linear networks. **a**: $Q$ (Eq. 3) and $\ell_{1,2}$ (Eq. 5) penalties, assuming negligible $\gamma$ for the $Q$ penalty. Log scale on both axes. **b**: Main task generalization loss for networks trained with MTL, PT+FT, single-task learning (STL), PT + finetuning via linear probing (PT+FT (LP)), and single-task linear probing (STL (LP), equivalent to lazy single-task learning, or ridge regression). Log scale on both axes. **c**: Proportion of variance concentrated in the top $k$ weights, as a function of $k$ (for an overlap of 30/40). The rapid decrease demonstrates the sparsity of the learned solution. **d**: Proportion of weight norm in the 40 dimensions relevant for the auxiliary task (again for an overlap of 30/40). **e**: Generalization loss for case in which auxiliary task (with 40 nonzero ground-truth weights) and main task (number of ground-truth weights indicated by color scale) share no common features. **f**: Generalization loss for case in which main task uses a subset of the features used by the auxiliary task. **g**: Same as PT+FT case in panel $e$, but with the network weights rescaled by 0.1 or 0.01 (panel title) following pretraining. A sparsity bias is evident, unlike in $e$ (rescaling = 1.0 case), and more pronounced as rescaling coefficient decreases. **h**: Unexplained variance as a function of weight scaling. On low numbers of samples, low scalings result in much more pronounced sparsity. **i**: Performance in the case of 5 main task features chosen either as a subset of the auxiliary task features ("shared") or disjoint from them ("task-specific"), varying the rescaling of weights following pretraining (1, 0.1, and 0.01). A bias toward feature reuse is evident even at the low scalings which yield a sparsity bias in panels $g$, $h$.

above, which coincide with the bias of single-task rich-regime (but not lazy-regime) learning. When the tasks partially overlap, MTL and PT+FT outperform both single-task learning and a PT + linear probing strategy (finetuning only the second-layer weights $w^{(2)}_{+,d}$ and $w^{(2)}_{-,d}$), which by construction cannot select task-specific features. Thus, both PT+FT and MTL are capable of simultaneously exhibiting a feature sharing bias while also displaying task-specific feature selection, consistent with the hybrid $\ell_1$ / weighted-$\ell_2$ regularization penalties derived above. Interestingly, PT+FT performs better than MTL when the tasks use identical feature sets. This behavior is consistent with the $Q$-norm more strongly penalizing new-feature learning than the MTL norm, as observed in Section 3.3.

To more directly test for a bias toward sparsity in task-specific features, we computed the fraction of overall weight norm in the learned main task linear predictor $\vec{\beta}$ that is captured by the top k strongest weights. We confirmed that the learned linear maps are indeed effectively sparse for both MTL and PT+FT, even when the main and auxiliary tasks contain distinct features and few samples

are available (Fig. 1c for 30/40 overlap case, see Appendix E, Fig. 5e for full suite of experiments)[2]. Further, to test for a bias toward feature sharing, we computed the fraction of the norm of $\vec{\beta}$ captured by the 40 features learned on the main task (Fig. 1d, see Appendix E, Fig. 5f for full suite of experiments). For MTL and PT+FT, this fraction is high for very few samples (indicating an inductive bias toward feature sharing) and gradually approaches the true overlap (30/40=0.75). Finally, we also directly measured the $\ell_{1,2}$ and $Q$ norms of the solutions learned by networks (Appendix E, Fig. 5a), confirming a bias toward minimization of these norms in MTL and PT+FT, respectively.

As another test of the predicted bias toward sparsity in task-specific features, we conducted experiments in which the main and auxiliary task features do not overlap, and varied the number $k_{main}$ of main task features. We find that both MTL and PT+FT are more sample-efficient when the main task is sparser, consistent with the prediction (Fig. 1e).

### 4.2 PT+FT EXHIBITS A SCALING-DEPENDENT NESTED FEATURE-SELECTION REGIME

In the limit of small $\frac{|\beta_d|}{|\beta_d^{aux}|}$, both the MTL and PT+FT penalties converge to weighted $\ell_2$ norms. Notably, the behavior is $\ell_2$-like even when $\frac{|\beta_d|}{|\beta_d^{aux}|} \approx 1$ (Fig. 1a). Thus, among features that are weighted as strongly in the auxiliary task as the main task, the theory predicts that PT+FT and MTL should exhibit no sparsity bias. To test this, we use a teacher-student setting in which all the main task features are a subset of the auxiliary task features, i.e. $k_{main} \leq k_{aux}$, and the number of overlapping units is equal to $k_{main}$. We find that MTL and PT+FT derive little to no sample efficiency benefit from sparsity in this context, consistent with an $\ell_2$-like minimization bias (Fig. 1f).

However, as remarked in Section 3.3, in the regime where $\frac{|\beta_d|}{|\beta_d^{aux}|}$ is greater than 1 but not astronomically large, the PT+FT penalty maintains an inverse dependence on $|\beta_d^{aux}|$ while exhibiting approximately $\ell_1$ scaling. In this regime, we would expect PT+FT to be adept at efficiently learning the tasks just considered, which require layering a bias toward sparse solutions on top of a bias toward features learned during pretraining. We can produce this behavior in these tasks by rescaling the weights of the network following pretraining by a factor less than 1. In line with the prediction of the theory, performing this manipulation enables PT+FT to leverage sparse structure *within* auxiliary task features (Fig. 1g). We confirm that weight rescaling does in fact lead to extraction of a sparse set of features by analyzing, as in Fig. 1c, the extent to which the learned linear predictor on the main task is concentrated on a small set of features (Fig. 1h). We also confirm that networks in the nested feature selection regime retain their ability to privilege features learned during pretraining above others (Fig. 1i), and that this phenomenon results from a bias toward feature reuse that grows less strong as the weight rescaling factor is decreased (Appendix 5, Fig. 5b).

This (initialization-dependent, $\ell_1$-minimizing) behavior is qualitatively distinct from the (initialization-dependent, weighted $\ell_2$-minimizing) lazy regime and the (initialization-independent, $\ell_1$-minimizing) feature-learning regimes. We refer to it as the *nested feature-selection* regime. This inductive bias may be useful when pretraining tasks are more general or complex (and thus involve more features) than the target task. This situation may be common in practice, as networks are often pre-trained on general-purpose tasks before finetuning for more specific applications.

## 5 NONLINEAR NETWORKS

### 5.1 SIMILARITIES TO LINEAR MODELS: FEATURE REUSE AND SPARSE FEATURE LEARNING

We now examine the extent to which our findings above apply to nonlinear models, focusing on single hidden-layer ReLU networks. We find that, as in the diagonal linear case, MTL and PT+FT effectively leverage feature reuse (outperforming single-task learning when tasks share features, Fig. 2a, right) and perform effective feature learning of task-specific features (nearly matching rich single-task learning and substantially outperforming lazy single-task learning when task features are not shared, Fig. 2a, left panel). Moreover, as in the linear theory, both effects can be exhibited simultaneously (Fig. 2a, middle panels). We also confirm that task-specific feature learning exhibits a sparsity bias (greater sample efficiency when non-shared main task features are sparse, Fig. 2b).

---

[2]Note that there is a slight non-monotonicity in learned solution sparsity as a function of number of main task samples; this is because of the discrepancy of L1 norm minimization and L0 "norm" minimization (sparsity maximization), see Appendix E Fig. 5c,d

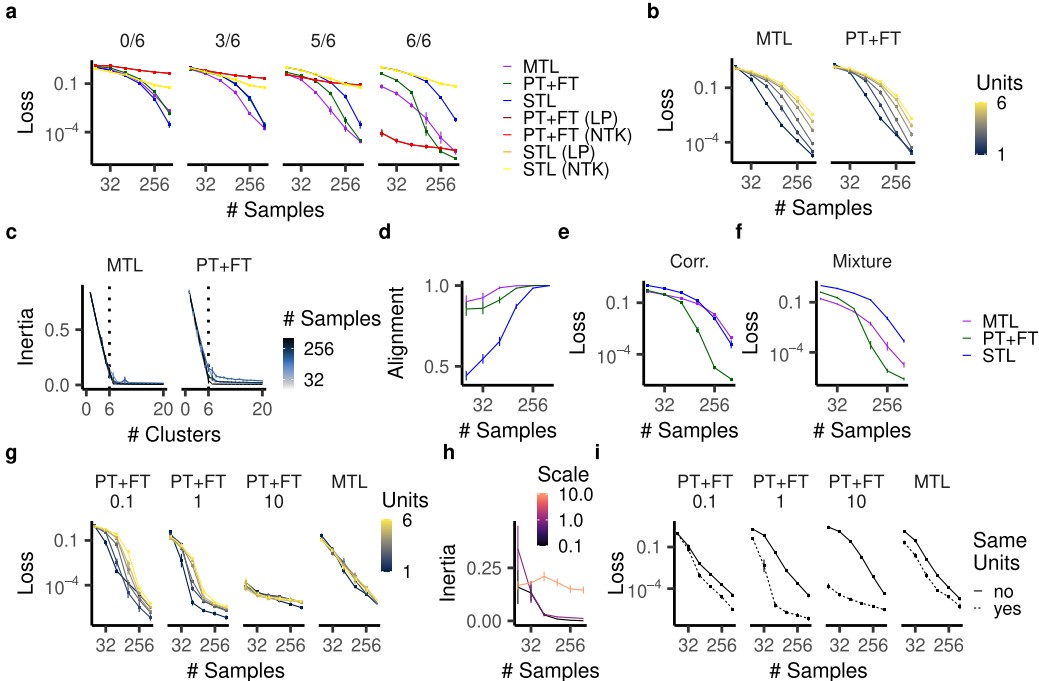

Figure 2: Nonlinear networks. **a**: Generalization loss (log-scaled) for different numbers of overlapping features (out of 6 total) between main and auxilliary tasks. NTK indicates the (lazy) tangent kernel solution. **b**: Generalization loss as a function of number of main task features (units in teacher network) in tasks where main and auxiliary task features are disjoint. **c**: Inertia (unexplained variance) for different numbers of clusters (overlap 5/6 case). The rapid decrease demonstrates the feature sparsity of learned solutions for both MTL and PT+FT. **d**: Alignment between the main task features and the best matching input-weight cluster found by k-means with K=12 (the total number of features for both tasks). The high alignment for PT+FT and MTL compared to STL demonstrates the bias toward feature sharing. **e**: Generalization loss in tasks where main-task features are correlated (0.9 cosine similarity of input weights) with corresponding auxiliary task features. **f**: Generalization loss for an example with both identically shared and correlated features between tasks. **g**, Generalization loss for PT+FT using different rescalings of network weights following pretraining (0.1, 1, and 10.0), and also for MTL, on tasks in which main task features are a subset of auxiliary task features. Color indicates number of main task features. **h**: Inertia for k-means clustering with a single cluster (K=1) for networks finetuned on a task with a single main task feature chosen from one of the auxiliary task features. Low inertia demonstrates that the network indeed learns a sparse solution. **i**: Generalization loss for same strategies as in panel *e*, on tasks in which main task features are either a subset of auxiliary task features ("shared") or disjoint ("task-specific").

We corroborate these claims by analyzing the sparsity of the learned solutions. We perform k-means clustering with $K$ clusters on the normalized input weights to each hidden-layer neuron in a network. We measure the extent to which $K$ cluster centers are able to explain the variance in input weights across hidden units; the fraction of variance left unexplained is commonly referred to as the "inertia." For values of $K$ at which the inertia is close to zero, we can say that (to a good approximation) the network effectively makes use of at most $K$ features. We find that the solutions learned by PT+FT and MTL are indeed quite sparse (comparable to the sparsity of solutions learned by single-task learning), even when the auxiliary task and main task features are disjoint (see Fig. 2c for representative example, and Appendix E, Fig. 6c,d for full suite of experiments), supporting the claim that PT+FT and MTL are biased toward sparsity in task-specific features. Further, the features learned by PT+FT and MTL are more aligned with the ground truth features than those learned by STL (Fig. 2d, see Appendix E, Fig. 6e for full suite of experiments), supporting the claim that PT+FT and MTL are biased toward sharing main and auxiliary task features.

## 5.2 PT+FT BIAS EXTENDS TO FEATURES CORRELATED WITH AUXILIARY TASK FEATURES

Interestingly, in cases with shared features between tasks, we find that finetuning can underperform multi-task learning (Fig. 2a), in contrast to the diagonal linear case. We hypothesize that this effect is caused by the fact that during finetuning, hidden units may not only change their magnitudes, but also the directions $\vec{\theta}_h$ of their input weights. Thus, in nonlinear networks, PT+FT may not strictly exhibit a bias toward reusing features across tasks, but rather a "softer" bias that also privileges features correlated with (but not identical to) those learned during pretraining. We conduct experiments in which the ground-truth auxiliary and main tasks rely on correlated but distinct features. Indeed, we find PT+FT outperforms MTL in this case (Fig. 2e). Thus, PT+FT (compared to MTL) trades off the flexibility to "softly" share features for reduced sample-efficiency when such flexibility is not needed. In Appendix E, Fig. 6e we show that MTL learns features that are more aligned with the ground-truth task features than PT+FT when main and auxiliary task features are identically shared, but the reverse is true when main and auxiliary task features are merely correlated.

In realistic settings, the degree of correlation between features across tasks is likely heterogeneous. To simulate such a scenario, we experiment with auxiliary and main tasks with a mixture of identically shared and correlated features. In this setting, we find that MTL outperforms PT+FT for fewer main task samples, while PT+FT outperforms MTL when more samples are available (Fig. 2f). We hypothesize that this effect arises because the flexibility of PT+FT to rotate hidden unit inputs is most detrimental in the few-sample regime where there is insufficient data to identify correct features.

## 5.3 THE NESTED FEATURE-SELECTION REGIME

In Section 4.2, we uncovered a "nested feature-selection" regime, obtained at intermediate values of $\frac{|\beta_d|}{|\beta_d^{aux}|}$ between the rich and lazy regimes, in which PT+FT is biased toward sparse feature selection biased by the features learned during pretraining. To test whether the same phenomenon arises in ReLU networks, we rescale the network weights following pretraining by various factors (which has the effect of scaling $|\beta_d^{aux}|$ for all $d$). We evaluate performance on a suite of tasks that vary the number of features in the main task teacher network and whether those features are shared with the auxiliary task. At intermediate rescaling values we confirm the presence of a nested feature selection regime, characterized by a bias toward sparsity among features reused from the auxiliary task (Fig. 2g) and a preference for reusing features over task-specific feature learning (Fig. 2i) which arises from a bias toward reusing auxiliary task features (Appendix E, Fig. 6g). Further rescaling in either direction uncovers the initialization-insensitive rich / feature-learning regime and the initialization-biased lazy learning regime. We do not observe nested feature selection behavior in MTL. Note that for different tasks and architectures, different rescaling values may be needed to enter the nested feature learning regime.

To shed further light on this regime, we analyzed the effective sparsity of learned solutions using the k-means clustering approach introduced previously. We find that networks identified above as in the nested feature selection regime indeed learn sparse (effectively 1-feature) solutions when the main task consists of a single auxiliary task feature (Fig. 2h). By contrast, networks with weights rescaled by a factor of 10.0 following pretraining exhibit no such nested sparsity bias (consistent with lazy-regime behavior). Additionally, supporting the idea that the nested feature selection regime maintains a bias toward feature reuse (Fig. 1g, Fig. 2f), we find that networks in this regime exhibit higher alignment of learned features with the ground-truth teacher network when the main task features are a subset of the auxiliary task features, compared to when they are disjoint from the auxiliary task features (Appendix E, Fig. 6g). This alignment benefit is mostly lost when networks are rescaled by a factor of 0.1 following pretraining (a signature of rich-regime-like behavior).

## 6 PRACTICAL APPLICATIONS TO DEEP NETWORKS AND REAL DATASETS

Our analysis has focused on shallow networks trained on synthetic tasks. To test the applicability of our insights, we conduct experiments with convolutional networks (ResNet-18) on a vision task (CIFAR-100), using classification of two image categories (randomly sampled for each training run) as the primary task and classification of the other 98 as the auxiliary task. As in our experiments above, MTL and PT+FT improve sample efficiency compared to single-task learning (Fig. 3a). Moreover, the results corroborate our findings in Section 5.2 that MTL performs better than PT+FT with fewer main task samples, while the reverse is true with more samples. A similar finding was made in Weller et al. (2022) in natural language processing tasks.

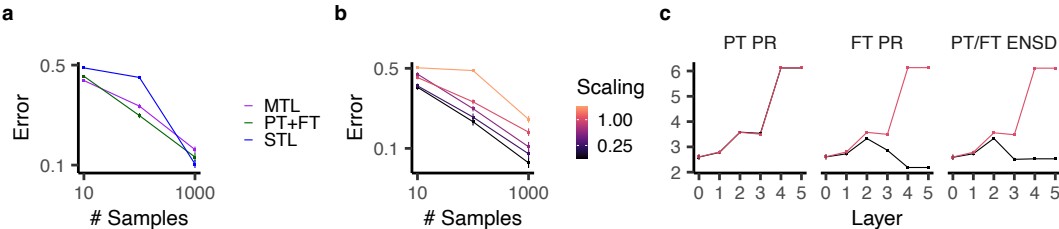

Figure 3: **a**: Test error on CIFAR-100 experiment as a function of main task samples (log scale on both axes). **b**: Test error on CIFAR-100 for PT+FT with different weight rescaling values following pretraining. **c**: Participation ratio (PR; measuring dimensionality) of the pretrained and finetuned networks and the effective number of shared dimensions (ENSD) between them.

Our findings in Section 4.2 and Section 5.3 indicate that the nested feature selection bias of PT+FT can be exposed or masked by rescaling the network weights following pretraining. Such a bias may be beneficial when the main task depends on a small subset of features learned during pretraining, as may often be the case in practice. We experiment with rescaling in our CIFAR setup. We find that rescaling values less than 1 improve finetuning performance (Fig. 3b). These results suggest that rescaling network weights before finetuning may be practically useful. We corroborate this hypothesis with additional experiments using networks pre-trained on ImageNet (see Appendix F).

To facilitate comparison of the phenomenology in deep networks with our teacher-student experiments above, we propose a signature of nested feature selection that can be characterized without knowledge of the underlying feature space. Specifically, we propose to measure (1) the *dimensionality* of the network representation pre- and post-finetuning, and (2) the extent to which the representational structure post-finetuning is shared with / inherited from that of the network following pretraining prior to finetuning. We employ the commonly used *participation ratio* (PR; Gao et al., 2017) as a measure of dimensionality, and the *effective number of shared dimensions* (ENSD) introduced by Giaffar et al. (2023), a soft measure of the number of aligned principal components between two representations. Intuitively, the PR and ENSD of network representations pre- and post-finetuning capture the key phenomena of the nested feature selection regime: we expect the dimensionality of network after finetuning to be lower than after pretraining ($PR(\mathbf{X}_{FT}) < PR(\mathbf{X}_{PT})$), and for nearly all of the representational dimensions expressed by the network post-finetuning to be inherited from the network state after pretraining ($ENSD(\mathbf{X}_{PT}, \mathbf{X}_{FT}) \approx PR(\mathbf{X}_{FT})$). We show that this description holds in our nonlinear teacher-student experiments with networks in the nested feature selection regime (rescaling factor 1.0) (Appendix G, Fig. 8c). Moreover, we find that the ResNet-18 model with rescaling applied (but not without rescaling) exhibits the same phenomenology (Fig. 3c). This supports the hypothesis that the observed benefits of rescaling indeed arise from pushing the network into the nested feature selection regime. See Appendix G for more details.

## 7 CONCLUSION

In this work we have provided a detailed characterization of the inductive biases associated with two common training strategies, MTL and PT+FTWe find that these biases incentivize a combination of feature sharing and sparse task-specific feature learning. In the case of PT+FT, we characterized a novel learning regime – the nested feature-selection regime – which encourages sparsity *within* features inherited from pretraining. This insight motivates simple techniques for improving PT+FT performance by pushing networks into this regime, which shows promising empirical results. We also identified another distinction between PT+FT and MTL – the ability to use "soft" feature sharing – that leads to a tradeoff in their relative performance as a function of dataset size.

More work is needed to test these phenomena in more complex tasks and larger models. There are also promising avenues for extending our theoretical work. First, in this paper we did not analytically describe the dynamics of PT+FT in ReLU networks, but we expect more progress could be made on this front. Second, our diagonal linear theory could be extended to the case of the widely used cross-entropy loss (see Appendix C for comments on this point). Third, we believe it is important to extend this theoretical framework to more complex network architectures. Nevertheless, our present work already provides new and practical insights into the function of auxiliary task learning.

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
