# A DERIVATION OF THE NORM MINIMIZATION BIAS OF PT+FT FOR DIAGONAL LINEAR NETWORKS

We provide a derivation of the norm minimization biases of diagonal linear networks; note that the same result is proved in Azulay et al. (2021).

Recall the parameterization of single-output diagonal linear networks $f : \mathbb{R}^d \to \mathbb{R}$:

$$f_w(\vec{x}) = \vec{\beta}(\vec{w}) \cdot \vec{x}, \qquad \vec{\beta}(\vec{w}) \in \mathbb{R}^D, \tag{6}$$

$$\beta_d(\vec{w}) := w_{+,d}^{(2)} \circ w_{+,d}^{(1)} - w_{-,d}^{(2)} \circ w_{-,d}^{(1)}, \tag{7}$$

where $\circ$ indicates elementwise multiplication.

We proceed by calculating the gradient flow dynamics of the task loss $L$ with respect to the weights $\vec{w}$. We adopt the notation and strategy of Woodworth et al. (2020). Using the notation $\tilde{X} = [X \ -X]$, we have:

$$\dot{\vec{w}}^{(1)}(t) = \nabla_{\vec{w}^{(1)}(t)} L = \nabla_{\vec{w}^{(1)}(t)} \left( \|\tilde{X}\left(\vec{w}^{(1)}(t) \circ \vec{w}^{(2)}(t)\right) - y\|_2^2 \right) = -2\tilde{X}^\top r(t) \circ \vec{w^{(2)}}(t) \tag{8}$$

$$\dot{\vec{w}}^{(2)}(t) = \nabla_{\vec{w}^{(2)}(t)} L = \nabla_{\vec{w}^{(2)}(t)} \left( \|\tilde{X}\left(\vec{w}^{(1)}(t) \circ \vec{w}^{(2)}(t)\right) - y\|_2^2 \right) = -2\tilde{X}^\top r(t) \circ \vec{w^{(1)}}(t) \tag{9}$$

If we change coordinates and define $\vec{s} = \frac{1}{2}\left(\vec{w}^{(1)} + \vec{w}^{(2)}\right)$ and $\vec{d} = \frac{1}{2}\left(\vec{w}^{(1)} - \vec{w}^{(2)}\right)$ we have:

$$\dot{\vec{s}}(t) = -2\tilde{X}^\top r(t) \circ \vec{s}(t), \tag{10}$$

$$\dot{\vec{d}}(t) = 2\tilde{X}^\top r(t) \circ \vec{d}(t). \tag{11}$$

$$\tag{12}$$

The solutions to these equations are

$$\vec{s}(t) = \vec{s}(0) \circ \exp\left(-2\tilde{X}^\top \int_0^t r(s)ds\right) \tag{13}$$

$$\vec{d}(t) = \vec{d}(0) \circ \exp\left(2\tilde{X}^\top \int_0^t r(s)ds\right) \tag{14}$$

We are interested in the form of the solution $\vec{\beta}$:

$$\vec{\beta}_{\vec{w}}(t) = \vec{w}_+^{(1)}(t)\vec{w}_+^{(2)}(t) - \vec{w}_-^{(1)}(t)\vec{w}_-^{(2)}(t), \tag{15}$$

$$= \left((\vec{s_+}(t))^2 - (\vec{d_+}(t))^2\right) - \left((\vec{s_-}(t))^2 - (\vec{d_-}(t))^2\right), \tag{16}$$

$$= \left((\vec{s_+}(t))^2 - (\vec{s_-}(t))^2\right) - \left((\vec{d_+}(t))^2 - (\vec{d_-}(t))^2\right). \tag{17}$$

We now make the assumption that the weights are initialized with $\vec{w}_+(0) = \vec{w}_-(0) = \vec{w}_0$ and define $\vec{s_0} = \frac{1}{2}\left(w_0^{(1)} + w_0^{(2)}\right)$ and $\vec{d_0} = \frac{1}{2}\left(w_0^{(1)} - w_0^{(2)}\right)$. Then,

$$\vec{\beta}_{\vec{w}}(t) = 2\vec{s_0}^2 \circ \left(\exp\left(-4X^\top \int_0^t r(s)ds\right) - \exp\left(4X^\top \int_0^t r(s)ds\right)\right) \tag{18}$$

$$-2\vec{d_0}^2 \circ \left(\exp\left(4X^\top \int_0^t r(s)ds\right) - \exp\left(-4X^\top \int_0^t r(s)ds\right)\right) \tag{19}$$

$$= 2(\vec{s_0}^2 + \vec{d_0}^2) \cdot \sinh\left(-4X^\top \int_0^t r(s)ds\right) \tag{20}$$

$$= \left(((w_0^{(1)})^2 + ((w_0^{(2)})^2\right) \cdot \sinh\left(-4X^\top \int_0^t r(s)ds\right) \tag{21}$$

Now the solution is in the same form equation (17) of in Appendix D of Woodworth et al. (2020), but with the coefficient in front of the sinh term replaced by $\left(((w_0^{(1)})^2 + ((w_0^{(2)})^2\right)$. Following the

rest of their argument, it follows that under the assumption that $\vec{\beta}(\infty)$ fits the training data with zero error, among all such solutions, $\vec{\beta}(\infty)$ minimizes the penalty

$$\|\beta_d\|_Q := \sum_{d=1}^{D} \left( ((w_0)_d^{(1)})^2 + ((w_0)_d^{(2)})^2 \right) q \left( \frac{2\beta_d}{((w_0)_d^{(1)})^2 + ((w_0)_d^{(2)})^2} \right), \tag{22}$$

$$q(z) = 2 - \sqrt{4 + z^2} + z \cdot \mathrm{arcsinh}(z/2) \tag{23}$$

If we assume that the initialization of $\vec{w_0}^{(2)}$ is a vector with constant entries $\gamma$ and that $\vec{w_0}^{(1)}$ is obtained from fitting solution to a pretraining task with effective weights $\vec{\beta}^{aux}$, the assumption of constant magnitude $\alpha$ in the initialization of the first and second-layer weights implies $(w_0)_d^{(1)} = \sqrt{|\beta_d^{aux}|}$, the result in Equation 3 follows.

# B    DERIVATION OF THE $\ell_{1,2}$ PENALTY MINIMIZATION BIAS OF MTL

## B.1    MULTI-TASK DIAGONAL LINEAR NETWORKS

Diagonal linear networks with $O$ outputs are parameterized as:

$$f_w(x) = \vec{\beta}(\vec{w})x, \qquad \vec{\beta}(\vec{w}) \in \mathbb{R}^{O \times D}, \tag{24}$$

$$\beta_{o,d}(\vec{w}) := w_{+,o,d}^{(2)} \circ w_{+,d}^{(1)} - w_{-,o,d}^{(2)} \circ w_{-,d}^{(1)}. \tag{25}$$

where $\circ$ indicates elementwise product. We are interested in how minimizing the total parameter norm

$$\sum_{d=1}^{D} \left( (w_{+,d}^{(1)})^2 + \sum_{o=1}^{O} (w_{+,o,d}^{(2)})^2 \right) + \left( (w_{-,d}^{(1)})^2 + \sum_{o=1}^{O} (w_{-,o,d}^{(2)})^2 \right) \tag{26}$$

maps to minimizing a norm over the solution weights $\vec{\beta}$. First we note that in the minimum parameter norm solution, for a given input dimension $d$, either all its associated $+$ weights or all its associated $-$ weights will be zero. Without loss of generality we may assume that all the $-$ weights are zero. So we are to minimize

$$\sum_{d=1}^{D} \left( (w_{+,d}^{(1)})^2 + \sum_{o=1}^{O} (w_{+,o,d}^{(2)})^2 \right) \tag{27}$$

For given solution coefficients $\vec{\beta}_d \in \mathbb{R}^O$, the value of the input weight $(w_{+,d}^{(1)})^2$ is a free parameter $z_d^2 > 0$, as we can set $w_{+,o,d}^{(2)} := \beta_{o,d}/z$. As any value for $z_d$ leads to the same function, we choose $z_d$ to minimize

$$((w_{+,d}^{(1)})^2 + \sum_{o=1}^{O} (w_{+,o,d}^{(2)})^2 = z_d^2 + \|\vec{\beta}_d\|_2^2/z_d^2. \tag{28}$$

Setting the derivative to zero implies

$$z_d^2 = \|\vec{\beta}_d\|_2. \tag{29}$$

As a result,

$$\sum_{d=1}^{D} \left( (w_{+,d}^{(1)})^2 + \sum_{o=1}^{O} (w_{+,o,d}^{(2)})^2 \right) = 2 \sum_{d=1}^{D} \|\vec{\beta}_d\|_2, \tag{30}$$

where the right-hand side of the equation is the $\ell_{1,2}$ norm.

Note that this implies, as a special case, that the minimum parameter solution to a diagonal linear network with one output minimizes the $\ell_1$ norm.

### B.2 Multi-task ReLU networks

Multi-task ReLU networks with a shared feature layer and $O$ outputs can be written as

$$f_w(x) = \sum_{h=1}^{H} \vec{w}_h^{(2)} (\langle \vec{w}_h^{(1)}, \vec{x} \rangle)_+ = \sum_{h=1}^{H} \vec{m}_h (\langle \vec{\theta}_h, \vec{x} \rangle)_+, \tag{31}$$

$$\vec{m}_h = \vec{w}_h^{(2)} \|\vec{w}_h^{(1)}\|_2, \vec{\theta}_h = \vec{w}_h^{(1)} / \|\vec{w}_h^{(1)}\|_2, \tag{32}$$

where $\vec{w}_h^{(2)}, \vec{m}_h \in \mathbb{R}^{\mathbb{O}}$. As above, we are interested in how minimizing the parameter norm $\sum_{h=1}^{H} \|\vec{w}_h^{(1)}\|_2^2 + \|\vec{w}_h^{(2)}\|_2^2$ maps to minimizing a norm over the solution weights $\vec{m}$. For a given $\vec{m}_h \in \mathbb{R}^O$, the norm of the input weight $\|\vec{w}_h^{(1)}\|_2$ is a free parameter $z_h > 0$, as we can set $\vec{w}_h^{(2)} := m_h/z$. As any value for $z_h$ leads to the same function, we choose $z_h$ so as to minimize

$$\|\vec{w}_h^{(1)}\|_2^2 + \|\vec{w}_h^{(2)}\|_2^2 = z_h^2 + \|\vec{m}_h\|_2^2 / z_h^2. \tag{33}$$

Setting the derivative to zero implies

$$z_h^2 = \|\vec{m}_h\|_2. \tag{34}$$

As a result,

$$\sum_{h=1}^{H} \|\vec{w}_h^{(1)}\|_2^2 + \|\vec{w}_h^{(2)}\|_2^2 = 2 \sum_{h=1}^{H} \|\vec{m}_h\|_2, \tag{35}$$

where the right-hand side of the equation is the $\ell_{1,2}$ norm.

## C Comments on cross-entropy vs. mean squared error loss

Cross-entropy and mean squared error are among the most common loss functions in machine learning. An important difference between them is that while mean squared error can be minimized to zero exactly by interpolating the data, cross-entropy achieves its minimum asymptotically as the model predictions become inifinitely large (Gunasekar et al., 2018). Consequently, while mean squared error is more amenable to an analysis of the full learning trajectory (Braun et al., 2022), cross-entropy is often more easily understood in the asymptotic limit of infinite training. For homogeneous networks, it has been shown that crossentropy induces an implicit regularization towards the minimal parameter $\ell_2$-norm (Lyu & Li, 2020; Nacson et al., 2019). Because all the networks we consider are homogeneous, by the results of Appendix B, diagonal neural networks trained on multiple tasks with crossentropy loss for infinite time would indeed minimize the $\ell_{1,2}$-norm, and ReLU networks would minimize the $\mathcal{F}_{1,2}$-norm. However, PT+FT networks would minimize the $\ell_1$ and $\mathcal{F}_1$ norms, respectively, behaving identically to single-task learning. This is because given infinite training time, the behavior of networks trained with cross-entropy loss is in theory independent of initialization. This behavior is quite different from that of networks trained with mean squared error loss at convergence, which is heavily dependent on initialization (indeed, this is the basis of our investigation in this work). However, prior work has shown that given finite training time, networks trained on cross-entropy loss learn solutions that are sensitive to initialization, and indeed there is a correspondence between increasing training time and decreasing initialization scale (incentivizing rich / feature-learning behavior). Thus, we expect our qualitative findings are applicable to the case of networks trained with cross-entropy loss for finite time. Moreover, our results on the effects of rescaling network parameters (e.g. to uncover the nested feature-selection regime) may be able to be replicated in the cross-entropy setting by scaling training time.

## D Robustness of main results to choice of number of auxiliary task samples and input dimension

To increase confidence that our main results are robust to the number of data samples used (1024 auxiliary task samples and up to 1024 main task samples in most of our experiments), and the number of ground-truth units in the teacher network (6), we repeated the experiments of Fig. 2a with 8192 auxiliary task samples and 40 ground-truth features. Indeed, in this setting the rich regime also helps with generalization if and only if the teacher units are sparse (Fig. 4a). Further, MTL and PT+FT tend to outperform STL if the features are overlapping and MTL tends to outperform PT+FT (Fig. 4b). In particular, the finetuned networks still benefit from feature learning, especially if some features are novel.

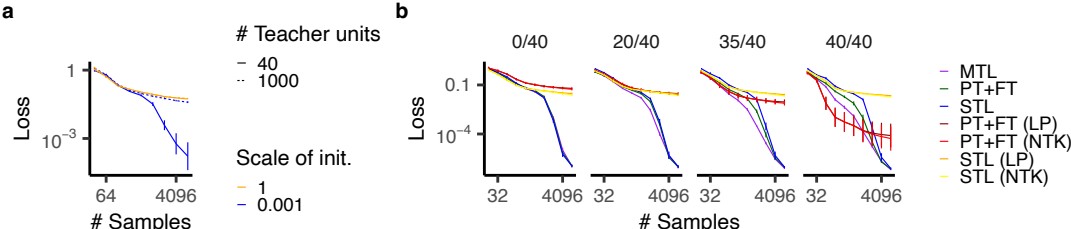

Figure 4: Larger-scale teacher-student experiments. **a**, Generalization loss of shallow ReLU networks trained on data from a ReLU teacher network (as in Fig. **??**, except with more teacher units and more data). **b**, Generalization loss for different numbers of overlapping features (out of 40 total) between main and auxiliary tasks. NTK indicates the (lazy) tangent kernel solution. This is comparable to Fig. 2a, except with more teacher units and more data.

## E ANALYSIS OF LEARNED SOLUTIONS IN LINEAR AND NONLINEAR NETWORKS

Our linear theory predicts inductive biases towards solutions that minimize norms, often either $\ell_1$-like (incentivizing sparsity) or $\ell_2$-like. Our experiments in Fig. 2 corroborate these description by analyzing how sample complexity depends on the feature sparsity of the ground-truth task solution, and how the sparse feature structures of the main and auxliary tasks relate. However, this evidence for sparsity biases (or lack thereof) is indirect; here we present more direct analyses of the learned solutions in linear and nonlinear networks that support the account we provide in the main text.

### E.1 DIAGONAL LINEAR NETWORKS

To check whether the implicit regularization theory is a good explanation for these performance results, we directly measured the $\ell_{1,2}$ and $Q$ norms of the solutions learned by networks, compared to the corresponding penalties of the ground truth weights. In Fig. 5a we see that as the amount of training data increases, the norms all converge to that of the ground truth solution, but in the low-sample regime, MTL and PT+FT find solutions with lower values of their corresponding norm than the ground-truth function, consistent with the implicit regularization picture (by contrast, STL does not consistently find solutions with lower values of these norms than the ground truth).

Our theory predicts that weight rescaling by a factor less than 1.0 following pretraining reduces the propensity of the network to share features between auxiliary and main tasks during finetuning. We confirm that this is the case in Fig. 5b by analyzing the overlap between the auxiliary task features and the learned linear predictor for the main task.

In Fig. 5c we show that our measure of effective sparsity of learned solutions in diagonal linear networks effectively distinguishes between networks trained in the feature selection regime and networks trained with linear probing (only training second-layer weights). Moreover, in Fig. **??** we show that the L1 norm of the solution increaeses with the training sample size, consistent with a bias towards L1 minimization. There is an interesting discrepancy between the behavior of the sparsity of the solutions (nonmonotonic, see Fig. 5c) and the L1 norm (largely monotonic, see Fig. 5d). This is attributable to the discrepancy between the L1 norm (which diagonal linear networks in the rich regime are biased to minimize) and sparsity (for which L1 norm is only a proxy).

In Fig. 5e we show the same information as Fig. 1c but for different values of the number of over-lapping features between main and auxiliary tasks (each of which uses 40 features). We find that, as in the example shown in the main text, learned solutions across a range of overlaps are as sparse as using single-task learning (see ig. 5c) when task features do not overlap (0/40 case) and more sparse otherwise (on account of the bias toward reuse of the sparse features learned during the auxiliary task, see next paragraph)..

In Fig. 5f we show the same information as Fig. 1d but for different values of the number of over-lapping features between main and auxiliary tasks (each of which uses 40 features). We find that, as in the example shown in the main text, learned main task solutions are biased to share auxiliary task features when few samples are available.

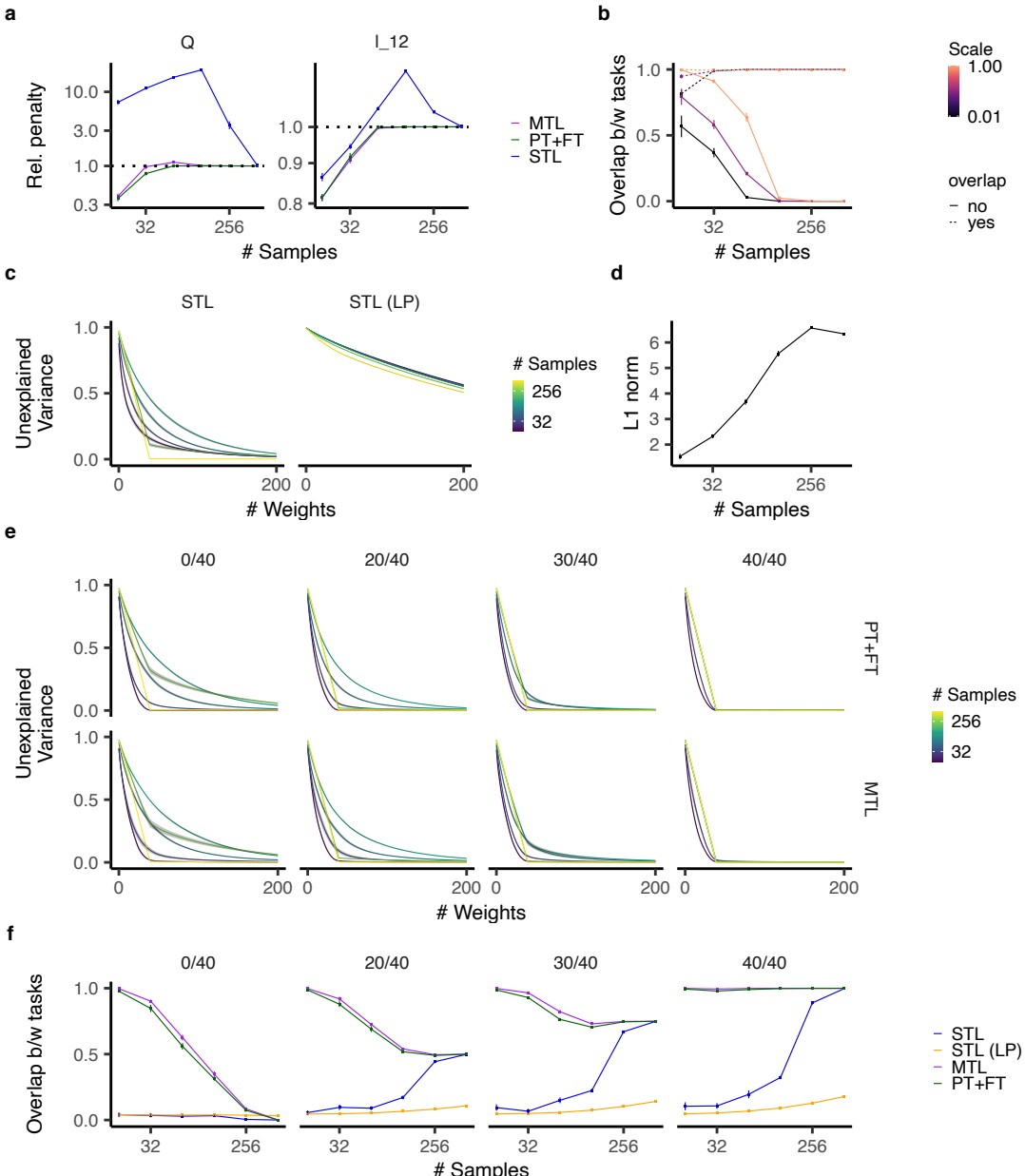

Figure 5: **a**, $\ell_{1,2}$ norm and $Q$ penalty for MTL, STL, and PT+FT networks from Fig. 1a (40/40 overlapping features case). **b**, Proportion of the weight norm in the 40 dimensions relevant for the auxiliary task, for the networks in Fig. 1i. Weight rescaling decreases this overlap. **c**, Proportion of variance concentrated in the top $k$ weights, as a function of $k$, for training on a single-task. When both layers are trained from small initialization (STL), this variance decreases much more rapidly than for pure linear readout training (STL (LP)), demonstrating the sparsity of the learned solution. **d**, L1 norm for STL as a function of the number of samples. **e**, Proportion of variance across different overlaps and for different learning setups (see also Fig. 1c). The rapid decrease in variance demonstrates the sparsity of the learned solutions both for PT+FT and MTL. **f**, Proportion of weight norm in the 40 dimensions relevant for the auxiliary task (see also Fig. 1d).

## E.2 ReLU networks

We adopt a clustering-based approach to analyzing the effective sparse structure of learned task solutions. Specifically, for a given trained network, we perform k-means clustering with a predetermined value of $K$ clusters on the normalized input weights to each hidden-layer neuron in the network[3]. We measure the extent to which $K$ cluster centers are able to explain the variance in input weights across hidden units; the fraction of variance left unexplained is commonly referred to as the "inertia." For values of $K$ at which the inertia is close to zero, we can say that (to a good approximation) the network effectively makes use of at most $K$ nonlinear features.

### E.2.1 Single-task learning: rich inductive bias yields clusters of similarly tuned neurons that approximate sparse ground-truth features

In the single-task learning case, we measure the inertia of trained networks in Fig. **??**d as a function of $K$. We find that for networks in the rich regime (small initialization scale), for tasks with sparse ground-truth (six units in the ReLU teacher network), the networks do indeed learn solutions that make use of approximately six nonlinear features (Fig. 6a). For tasks with many (1000) units in the teacher network, the network finds solutions that use a small number of feature clusters when main task samples are limited, but gradually uses more clusters as the number of samples is increased (Fig. 6a), at which point the network matches the teacher function very well, see Fig. **??**d). This bias towards sparser-than-ground-truth solutions given insufficient data corroborates our claim of an inductive bias towards sparse solutions. By contrast, networks in the lazy learning regime (large initialization scale) display no such bias, corroborating our claim that the sparse $\ell_1$-like inductive bias is a property of the rich regime but not the lazy regime. Interestingly, in the sparse ground-truth case learned solutions are relatively less sparse for an intermediate number of training examples. This may arise because an $\ell_1$-like inductive bias is not exactly the same as a bias toward sparse solutions over nonlinear features, particularly when training data is limited. We leave an in-depth investigation of this phenomenon to future work.

Our clustering analysis allows us to measure the extent to which the effective features employed by the network (cluster centers) are aligned with the ground-truth task features. Specifically, for each teacher unit, we compute an "alignment score" between teacher and student networks by taking each teacher unit, measuring its cosine similarity with all the cluster centers, choosing the maximum value, and averaging this quantity across all teacher units. We find that the learned feature clusters are indeed highly aligned with the ground-truth teacher features in the sparse ground-truth case, and moreso as the number of main task samples (and consequently task performance) increases (Fig. 6b).

### E.2.2 Pretraining+finetuning finds sparse solutions and improves alignment of feature clusters learned during pretraining

We find that pretraining+finetuning improves performance over single-task learning when main and auxiliary task features are shared (or correlated), and maintains an apparent bias toward sparsity in new task-specific features. To corroborate these claims, we performed our clustering analysis on the solutions learned through PT+FT. We find that the solutions learned are indeed quite sparse (comparable to the sparsity of solutions learned by single-task learning), even when the auxiliary task and main task features are disjoint (Fig. 6c). Moreover, we find that MTL also learns sparse solutions (Fig. 6d). In particular, as expected, the effective features on tasks with overlapping features is equal to the number of total unique features. Moreover, we observe that when main task and auxiliary task features are shared, PT+FT and MTL networks exhibit higher alignment between learned features and ground-truth features than single-task-trained networks, especially when main task samples are limited (Fig. 6e). This provides a mechanistic underpinning for the relationship between the inductive bias of PT+FT that we describe in the main text and its performance benefits.

---

[3]weighting the importance of each unit to the k-means objective by the weight of its contribution to the network's input-output function, specifically the magnitude of the product of its associated input and output weights

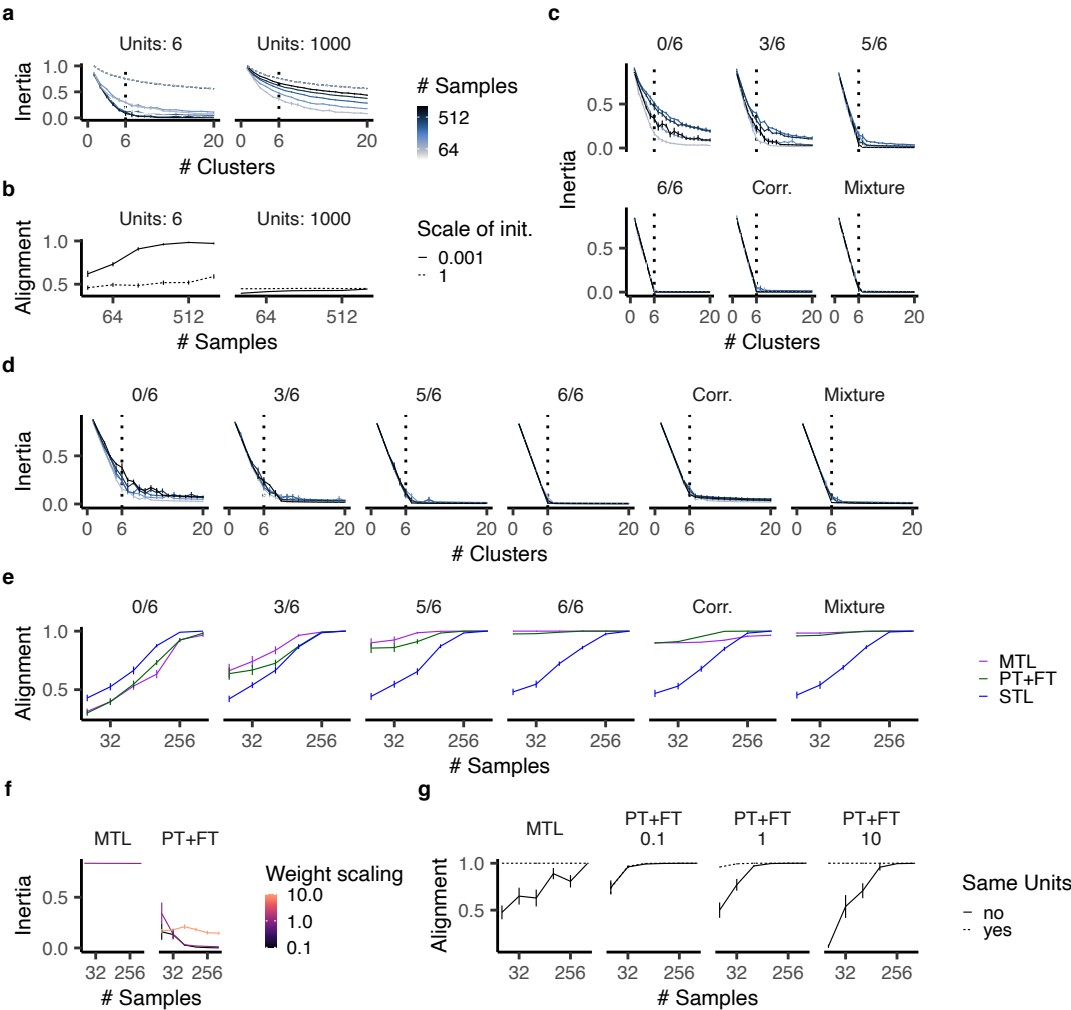

Figure 6: Analysis of effective sparsity of learned ReLU network solutions. **a** Inertia (k-means reconstruction error for clustering of hidden-unit input weights) as a function of the number of clusters used for $k$-means, for different numbers of main task samples and ground-truth teacher network units, in single-task learning. **b** Alignment score – average alignment (across teacher units) of the best-aligned student network cluster uncovered via k-means. **c**, Inertia for networks trained using PT+FT for the tasks of Fig. 2a and Fig. 2c. **d**, Same as panel $c$ but for networks trained with MTL. **e**, Alignment score for networks trained with MTL, PT+FT, and STL on the same tasks as in panels $c$ and $d$. **f** Inertia (using $k = 1$ clusters) for networks trained on an auxiliary task that relies on only one ground-truth feature, which is one of the six ground-truth features used in the auxiliary task (as in Fig. 2e,f), using MTL or PT+FT with various rescaling factors applied to the weights prior to finetuning. **g** Alignment score for the networks and task in panel $f$.

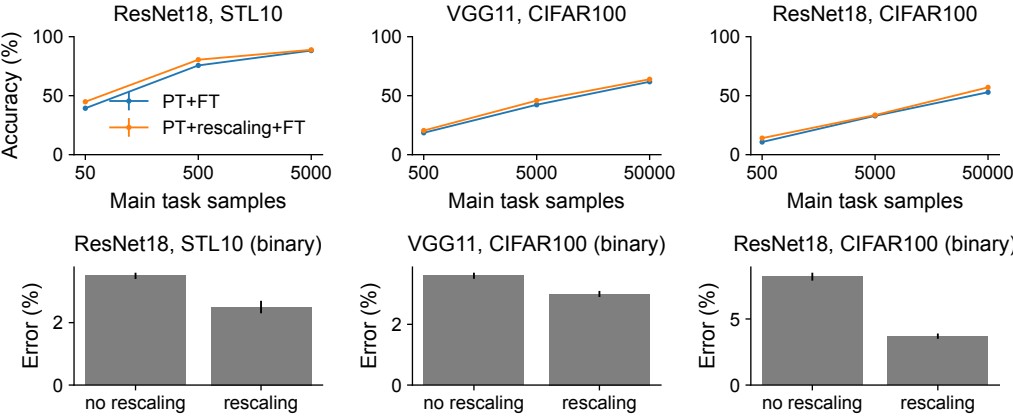

Figure 7: Results for finetuning deep convolutional networks trained on ImageNet, with/without weight rescaling (factor of 0.5) prior to finetuning.

### E.2.3 NESTED FEATURE SELECTION REGIME ALLOWS NETWORK TO PRIORITIZE A SPARSE SUBSET OF FEATURE CLUSTERS LEARNED DURING PRETRAINING

In the main text we describe the "nested feature selection" regime, which occurs at intermediate values of the ratio between ground-truth main task feature coefficients and pretrained network weight scale. In this regime, networks can more efficiently learn main tasks that make use of a subset of the features used in the auxiliary task (Fig. 1f, Fig. 2e) while still maintaining a bias towards reusing features from the auxiliary task (Fig. 1g, Fig. 2f). Here we show that networks in this regime (obtained most clearly in the shallow ReLU network case when networks are rescaled by a value of 1.0 after pretraining, see Fig. 2e,f) indeed learn very sparse (effectively 1-feature) solutions when the ground-truth main task consists of a single auxiliary task features (Fig. 6e, right). By contrast, networks with weights rescaled by a factor of 10.0 following pretraining exhibit no such nested sparsity bias (consistent with lazy-regime behavior). Similarly, multi-task networks cannot exhibit such a bias in their internal representation as they still need to maintain the features needed for the main task (Fig. 6e, left). Additionally, supporting the idea that the nested feature selection regime allows networks to benefit from feature reuse (Fig. 1i, Fig. 2i), we find that networks in this regime exhibit a higher alignment score with the ground-truth teacher network when the main task features are a subset of the auxiliary task features compared to when they are disjoint from the auxiliary task features (Fig. 6g). This alignment benefit is mostly lost when networks are rescaled by a factor of 0.1 following pretrainning (a signature of rich-regime-like behavior).

## F FURTHER EVALUATIONS OF THE RESCALING METHOD FOR FINETUNING

To evaluate the robustness / general-purpose utility of our suggested approach of rescaling network weights following pretraining, we experimented with finetuning convolutional networks pretrained on ImageNet on downstream classification tasks: pretrained ResNet-18 finetuned on CIFAR100, pretrained VGG11 finetuned on CIFAR100, and pretrained ResNet-18 finetuned on STL-10. We experimented both with finetuning on the full multi-way classification task, and also on binary classification tasks obtained by subsampling pairs of classes from the main task dataset (which we found exposes performance differences more strongly). Due to computational constraints, we did not sweep over the choice of the rescaling factor, but simply used a factor of 0.5 in all cases. We find that rescaling improves finetuning performance, to varying degrees, in all of our experiments (Fig. 7).

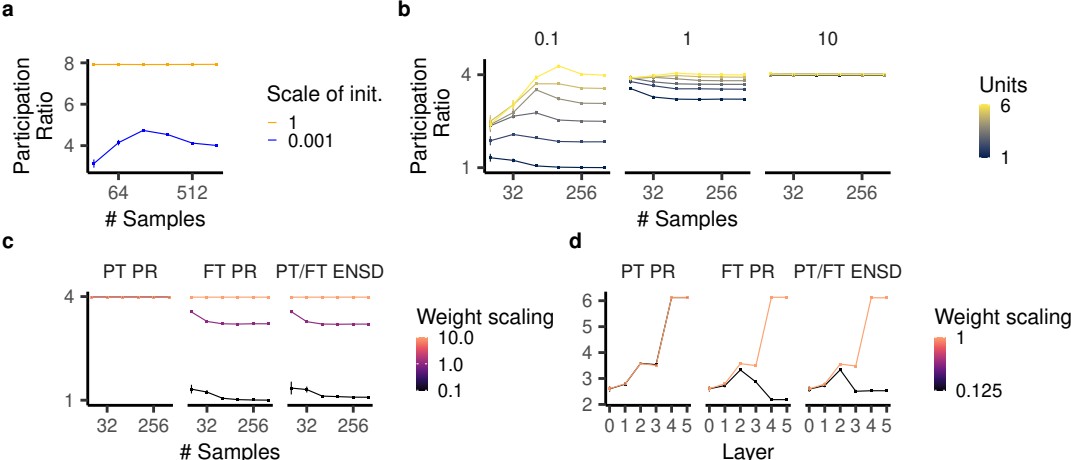

Figure 8: Dimensionality of the network representations before and after finetuning. **a**, Participation ratio of the ReLU networks' internal representation after training on a task with six teacher units. **b**, Participation ratio of the network representation after finetuning on the nested sparsity task with different weight rescalings. **c**, Participation ratio before (left panel) and after finetuning (middle panel) and the effective number of shared features between the two representations. Small weight scaling decreases the participation ratio after training. **d**, The same quantities for ResNet18 before and after finetuning (see also Fig. 2h).

## G    ANALYSIS OF REPRESENTATIONS LEARNED IN THE NESTED FEATURE SELECTION REGIME: BRIDGING THE GAP FROM SHALLOW TO DEEP NETWORKS

We were interested in testing whether our theoretical understanding of shallow networks is truly responsible for the behavior of deeper networks (with more direct evidence than performance results / sample complexity). Specifically, we sought to understand whether the observed benefit of rescaling network weights following pretraining (Fig. 2h, Appendix. F) relates to the nested feature selection regime we characterized in shallow networks. Doing so is challenging, as the space of "features" learnable by a deep network is difficult to characterize explicitly (making the feature clustering analysis employed in Appendix. E inapplicable). To circumvent this issue, we propose a signature of nested feature selection that can be characterized without knowledge of the underlying feature space. Specifically, we propose to measure (1) the *dimensionality* of the network representation pre- and post-finetuning, and (2) the extent to which the representational structure post-finetuning is shared with / inherited from that of the network following pretraining prior to finetuning.

We employ the commonly used *participation ratio* (PR; Gao et al., 2017) as a measure of dimensionality. For an $n \times p$ matrix $\mathbf{X}$ representing $n$ mean-centered samples of p-dimensional network responses, with a $p \times p$ covariance matrix $\mathbf{C}_X = \frac{1}{n}\mathbf{X}^\top\mathbf{X}$, the participation ratio is defined as

$$PR(X) = \frac{\left(\sum_{i=1}^{p} \lambda_i\right)^2}{\sum_{i=1}^{p} \lambda_i^2} = \frac{trace\left(\mathbf{C}_X\right)^2}{trace\left(\mathbf{C}_X^2\right)} = \frac{trace\left(\mathbf{X}^\top\mathbf{X}\right)^2}{trace\left(\mathbf{X}^\top\mathbf{X}\mathbf{X}^\top\mathbf{X}\right)}$$

where $\lambda_i$ are the eigenvalues of the covariance matrix $\mathbf{C}_X$. The PR scales from 1 to $p$ and measures the extent to which the covariance structure of responses $\mathbf{X}$ is dominated by a few principal components or is spread across many. We argue that low-dimensional representations are a signature of networks that use a sparse set of features. We confirm that this is the case in our teacher-student setting: networks in the rich regime, which are biased towards sparse solutions, learn representations with lower PR than networks in the lazy regime, which are not biased toward sparse solutions (Fig. 8a).

Our measure of shared dimensionality between two representations is the *effective number of shared dimensions* (ENSD) introduced by Giaffar et al. (2023). The ENSD for an $n \times p$ matrix of responses $\mathbf{X}$ from one network and an $n \times p$ matrix of responses $\mathbf{Y}$ from another network is given by

$$ENSD(X,Y) = \frac{trace\left(\mathbf{Y}^\top \mathbf{X}\mathbf{X}^\top \mathbf{Y}\right) \cdot trace\left(\mathbf{X}^\top \mathbf{X}\right) \cdot trace\left(\mathbf{Y}^\top \mathbf{Y}\right)}{trace\left(\mathbf{X}^\top \mathbf{X}\mathbf{X}^\top \mathbf{X}\right) \cdot trace\left(\mathbf{Y}^\top \mathbf{Y}\mathbf{Y}^\top \mathbf{Y}\right)}$$

This measure is equal to the centered kernel alignment (CKA), a measure of similarity of two network representations (Kornblith et al., 2019a), multiplied by the geometric mean of the participation ratios of the two representations. It measures an intuitive notion of "shared dimensions" — for example, if $\mathbf{X}$ consists of 10 uncorrelated units, if $\mathbf{Y}$ is taken from a subset of five of those units, the ENSD(X, Y) will be 5. If $\mathbf{Y}$ is taken to be five uncorrelated units that are themselves uncorrelated with all those in $\mathbf{X}$, the ENSD(X, Y) will be zero.

Intuitively, the PR and ENSD of network representations pre- and post-finetuning capture the key phenomena of the nested feature selection regime. In a case in which the main task uses a subset of the features of the auxiliary task, if the network truly extracts this sparse subset of features, we expect the dimensionality of network after finetuning to be lower than after pretraining ($PR(\mathbf{X}_{FT}) < PR(\mathbf{X}_{PT})$), and for nearly all of the representational dimensions expressed by the network post-finetuning to be inherited from the network state after pretraining ($ENSD(\mathbf{X}_{PT}, \mathbf{X}_{FT}) \approx PR(\mathbf{X}_{FT})$). By contrast, networks not in the nested feature selection regime should exhibit an $\ell_2$-like rather than $\ell_1$-like bias with respect to features inherited from pretraining and thus not exhibit a substantial decrease in dimensionality during finetuning.

We show that this description holds in our nonlinear teacher-student experiments. Networks that we identified as being in the "nested feature selection" regime (weights rescaled by 1.0 following pretraining), and also networks in the rich regime, exhibit decreased PR following finetuning (Fig. 8b). By contrast, lazy networks (weights rescaled by 10.0 following pretraining) exhibit no dimensionality decrease during finetuning. Additionally (see Fig. 8c), the ENSD between pretrained (PT) and finetuned (FT) networks is almost identical to the dimensionality of the finetuned representation (PR FT).

Strikingly, we observe very similar behavior in our ResNet-18 model pretrained on 98 CIFAR-100 classes and finetuned on the 2 remaining classes (Fig. 8d), when we apply our method of rescaling weights post-finetuning. Analyzing the PR and ENSD of the outputs of different stages of the network following pretraining and following finetuning, we see that dimensionality decreases with finetuning, and ENSD between the pretrained and finetuned networks is very close to the PR of the finetuned network. Moreover, this phenomenology is only observed when we apply the weight rescaling method; finetuning the raw pretrained network yields no dimensionality decrease. These results suggest that the success of our rescaling method may indeed be attributable to pushing the network into the nested feature selection regime.