# OpenReview forum: "Implicit regularization of multi-task learning and finetuning in overparameterized neural networks"
_ICLR.cc/2024/Conference — Submitted to ICLR 2024_

### Official Review · Reviewer_bJHA · 2023-10-25

**Soundness:** 2 fair
**Presentation:** 2 fair
**Contribution:** 1 poor
**Rating:** 3
**Confidence:** 4

**Summary:**

For two-layer diagonal linear networks, the current paper studies the implicit regularization of finetuning and multi-task learning. It derives norm-based measures that are implicitly minimized, and interprets them as inducing reuse of the auxiliary task weights and sparsity.

In the context of finetuning, the implicit regularization can be seen as an interpolation between L1 (rich) and L2 (kernel) regularizations, which under suitable parameter scaling leads to a form of “nested feature-learning,” whereby finetuning extracts salient weights from those learned by the pretrained model. Motivated by this observation, a heuristic of scaling down pretraining model parameters is suggested.

Experiments with shallow diagonal linear neural networks and ReLU networks corroborate the theoretical analysis.

**Strengths:**

The implicit regularization of finetuning and multi-task learning is a timely subject, given the popularity of these paradigms in practice compared to their limited formal understanding. Furthermore, despite their simplicity, two-layer diagonal linear networks exhibit non-trivial optimization dynamics, and have therefore been the focus of numerous studies. As such, I believe there is potential in investigating the implicit regularization brought forth by finetuning and multi-task learning for these models.

Unfortunately, the paper suffers from major issues in terms of novelty and significance of its contributions, as well as clarity of presentation. I therefore believe that it falls below the bar of acceptance. My concerns are specified in the weaknesses portion of the review below.

**Weaknesses:**

**Novelty and Significance:** The main contributions of Sections 4.1 and 4.2 are slight variations of known results. In particular, the characterization for the finetuning setting (Section 4.1) is a slight variation of Theorem 1 in [1], and the characterization for multi-task learning (Section 4.2) is a slight variation of Theorem 2 in [2]. Thus, beyond existing knowledge, the paper mostly provides interpretations of the sparsity inducing implicit regularization for diagonal linear neural networks. Such interpretations can be valuable, yet the setting considered is quite limited:

1. The analysis disregards the role of the auxiliary task data and objective, which are crucial in practice. In fact, the considered multi-task setting is equivalent to a single task with multiple outputs. Hence, the analysis inherently cannot provide insight as to how the implicit regularization of training with auxiliary tasks differs from single task training. Due to the substantial overlap between the results of Section 4 and prior work, I suggest as a promising direction forward to investigate how the effects of auxiliary task data and objective come into play.

2. The comparison between the implicit regularization of finetuning and multi-task learning is not an “apples to apples” comparison. For finetuning, the implicit regularizer is derived for gradient flow with the square loss, whereas for multi-task learning the L2 representation cost is considered, although it is not clear whether gradient descent/flow indeed leads to minimal L2 norm solutions in this case.

With regards to the empirical evaluation:

1. I find the empirical demonstration of the proposed heuristic (scaling down weights) to be insufficient. The evaluation includes only a single model (ResNet18) and a single dataset (variant of CIFAR100) with an oddly small number of training samples. Although the paper is mostly of theoretical nature, I still believe this does not meet the extent of evaluation expected when proposing a practical guideline.

2. Claims of an implicit regularization towards sparsity are measured in experiments only through sample complexity, as observed when plotting the loss against the number of samples. It would be more informative to measure the sparsity levels directly.

**Presentation:** The presentation severely lacks clarity. Claims are made without laying out their exact statements and under which assumptions they are proven. Specifically, in Section 4.1 it is not specified what the minimized objective is, what the optimization algorithm is, nor what the auxiliary weights stand for and how they are obtained. The appendix does not provide much clarity either, though after close examination it seems to prove a slight variation of Theorem 1 in Azulay et al. 2021 for the implicit regularization of diagonal linear networks optimized with gradient flow from a certain initialization.

**Additional Comments:**

1. A contributing factor to the lack of clarity is the use of undefined terminology, whose meaning is not obvious (at least to me). For example, what does “read-out weights” refer to in the context of diagonal linear neural networks? Furthermore, what does “linear probing” refer to? An additional fully connected linear layer? Training only $w^2_+$ and $w^2_-$?

2. In my opinion, the use of “feature-learning” is unconventional and can be misleading. Feature learning typically refers to learned representations based on which prediction is done, whereas here it refers to sparsity patterns in the weights of a linear predictor. Linear predictors are usually the prime example of a model that does not have a feature learning component. Perhaps “feature selection” is a more suitable term?

3. The reference to Lyu & Li, 2020 at the beginning of Section 4.2 is imprecise. They showed that for homogeneous neural networks, which include diagonal linear networks, convergence is in direction to a KKT point of a max-margin objective. This does not necessarily imply convergence to the exact minimizer. Theorem 4 in [3] does prove it though under some technical conditions.


[1] Azulay, Shahar, Edward Moroshko, Mor Shpigel Nacson, Blake E. Woodworth, Nathan Srebro, Amir Globerson, and Daniel Soudry. "On the implicit bias of initialization shape: Beyond infinitesimal mirror descent." In International Conference on Machine Learning, pp. 468-477. PMLR, 2021.

[2] Dai, Zhen, Mina Karzand, and Nathan Srebro. "Representation costs of linear neural networks: Analysis and design." Advances in Neural Information Processing Systems 34 (2021): 26884-26896.

[3] Gunasekar, Suriya, Jason D. Lee, Daniel Soudry, and Nati Srebro. "Implicit bias of gradient descent on linear convolutional networks." Advances in neural information processing systems 31 (2018).

**Questions:**

\-

---

> ### Author Response · Authors · 2023-11-16
> **Response to reviewer bJHA**
>
> We thank the reviewer for the thoughtful criticisms of our work and suggestions for improvement.  We have updated our manuscript in response to several of the reviewer’s points (major revisions for clarity are highlighted in red) and provide responses below. If the reviewer has further suggestions or comments we would be happy to address them during the discussion period.
>
>
> —  **Regarding our theoretical contribution**: we would like to emphasize that we do not believe the primary contribution of our paper is in proving new theorems.  As the reviewer notes, our main theoretical claims are applications/interpretations of existing results in the setting of pretraining+finetuning and multi-task learning; we have endeavored to make this more clear in the revised manuscript.  However, we believe that applying this theoretical framework to PT+FT and MTL is novel, and the insights gained are non-obvious and will be of interest to the community.  We would like to highlight some of these insights here.  (1) The intermediate regime between rich and lazy learning that we highlight (i.e. “nested feature selection”) is not emphasized in prior work, but we show is key to understanding the inductive biases of PT+FT.  (2) Our thorough suite of teacher-student experiments demonstrates consequences of L1/L2-like norm biases which may not be obvious — for instance, the fact that models can display a “rich” bias toward sparsity in new features learned during finetuning while maintaining a “lazy” lack of bias toward sparsity in use of features inherited from pretraining. (3) One of our key findings, not addressed in this review, is not captured at all by the linear theory, as it arises only in the nonlinear case due to the possibility of tasks involving correlated-but-not-identical nonlinear features.
>
> – **Regarding clarity:** we thank the reviewer for the helpful suggestions for improved clarity in the presentation of theoretical results and notation, which we have tried to address in the revised manuscript (see red text).  We have also changed “feature-learning” to “feature selection” where appropriate (i.e. in the linear setting), as suggested.
>
> – **Additional evidence for sparsity-incentivizing implicit regularization:** we agree that our paper would benefit from a more direct characterization of sparsity in learned solutions, particularly in the nonlinear networks where the applicability of the theory is less assured.  We have provided additional analyses that corroborate this description in the revised manuscript, provided in Appendix E.  See the “response to all reviewers” for a description of the key results of this analysis.
>
> – **Additional results for deep networks / realistic tasks:** We agree with the reviewer’s requests for more convincing empirical results in realistic networks/tasks and have done our best to meet them. We scaled up our CIFAR-100 experiment to use more samples.  See the “response to all reviewers” for a description of additional validation of the proposed rescaling method with different architectures and tasks (Appendix F).
>
> – **Additional analysis of nested feature selection regime in deep networks**: Please also see our “response to all reviewers” for description of an additional analysis (Appendix G) that provides evidence that the weight rescaling method does indeed cause the ResNet-18 network to display a particular signature of “nested feature learning” behavior during finetuning (which is also observed in the appropriately scaled shallow teacher-student networks).
>
> – **Regarding investigating the role of auxiliary task data / objective:** We agree this is an interesting and worthwhile direction for future work and are excited to pursue it.  However, we also think that our existing contributions constitute a paper’s worth of results on their own.
>
> – **Regarding the PT+FT / MTL comparison:** we agree that the theoretical results are not directly comparable and have further highlighted this in the revised manuscript.  However, we believe our teacher-student experiments demonstrate that the norms we derive for PT+FT (under gradient flow) and MTL (under L2 minimization) are useful descriptions in that they make accurate predictions about the relative performance of the two methods across a diverse suite of task setups.

---

> ### Comment · Reviewer_bJHA · 2023-11-19
>
> I have read the author's response and the other reviews carefully.
>
> 1. I acknowledge that, by the author's response, the theoretical analyses of Section 4 were not intended as main contributions. However, the paper is certainly not positioned accordingly. The abstract and introduction specify the description of solutions learned by two-layer diagonal linear networks as part of the main contributions. The fact that these are slight variations of known results is not mentioned in either places, nor in the related work section. For the finetuning implicit bias analysis this is only mentioned in the proof appendix, and for the multi-task setting it is only specified briefly in parenthesis. I find this to be misleading. The relation to prior work needs to be abundantly clear.
>
> &nbsp;
>
> 2. Given that the theoretical characterizations are not a main contribution, rather the insights and empirical analysis, the bar of empirical evidence becomes higher. The additional experiments in Appendix E are a good step forward, and should probably take a more central role given that they are the only direct support for the implicit bias towards sparsity claims.
>
>     Furthermore, the experiments in Appendix E consider only ReLU networks. To better substantiate the described nested feature selection and sparsity bias in task-specific weights, it would be beneficial to consider diagonal linear networks as well, in which they can be evaluated directly.
>
> &nbsp;
>
> 3. Presentation: Although improved, the statement of the theoretical results (Section 4) is still not formal. I strongly recommend abiding to the standard practice of stating formal results in a Theorem/Proposition/Corollary environment from which it should be clear what exactly is proven and under which conditions.
>
> &nbsp;
>
> Overall, I believe that the insights derived from existing theory, accompanied by empirical evidence, can be of interest and merit publication. However, in my opinion, the paper requires rather major modifications that will clarify its main contributions with respect to prior works, and improve its focus on the novel contributions (e.g. substantiating claims of the nested feature selection and task-specific weight sparsity). Unfortunately, I do not believe that such a revision is feasible as part of the author-reviewer discussion period.

---

> ### Author Response · Authors · 2023-11-23
> **Second response to reviewer bJHA**
>
> We thank the reviewer for participating in the discussion.  We sincerely appreciate the reviewer’s suggestions and attention to detail; we have updated our manuscript to address these additional comments (sections that have been revised or added are highlighted in blue font) and believe it has made our paper stronger and clarified its contributions.  At the same time, we wish to challenge some of the statements the reviewer has made about the work, which we believe overlook key aspects of the manuscript.
>
> We believe that our work provides several insights into the subject of multi-task learning and finetuning that are entirely novel and are of interest to the community.  As such, we argue that this work merits timely conference publication if the claims are technically sound and communicated accurately.  We would greatly appreciate if the reviewer could reassess the revised manuscript, which we believe meets this bar.
>
> Specific responses / comments addressed:
>
> – We have updated the abstract, introduction, and related work section to make more clear the close relationship between the theoretical statements in our work and those in prior work on diagonal linear networks.
>
> – We have rewritten the main theoretical claims more clearly and formally (see Corollary 1 and Corollary 2 in the revised draft)
>
> – The reviewer suggests that the empirical evidence for our key claims (in particular the claim of task-specific sparsity bias for PT+FT and MTL and the nested feature selection regime for PT+FT) is inadequate.  We respectfully disagree; the empirical evidence we provided in our initial submission is a direct demonstration of these key claims.  In particular, the reviewer indicates that the sample complexity plots in Figures 1 and 2 (formerly Figures 2 and 3) are only indirect evidence for the implicit penalties we claim for PT+FT and MTL.  However, the sample complexity results are in fact strong evidence, given our intentional selection of teacher-student experiments.  For instance, the improved sample complexity for sparser ground-truth in the case of non-overlapping main and auxiliary task features (Fig 1d and 2b) demonstrate an inductive bias towards sparsity in task-specific features.  Likewise, the improved sample complexity (or lack thereof) for sparser ground-truth features in the case in which main task features are a subset of auxiliary task features (Fig. 1e+f, 2e) demonstrates the presence (or absence) of the nested feature selection regime.  We also would like to draw attention to Fig. 1c (present since the initial submission), which provides direct evidence for the implicit regularization claims in the diagonal linear case (which we have now moved to Appendix E, Fig. 5a to make room for the sparsity analysis requested).
>
> – That said, we agree with the reviewer that a direct analysis of the sparsity of learned solutions is helpful and complements our existing analyses.  We have integrated the highlights of the analysis of sparsity in solutions in the nonlinear case (Appendix E)  into the main text (Fig. 2).  Moreover, as requested, we have conducted a similar analysis in the diagonal linear case, the highlights of which are now included in Fig. 1 (and more details on the linear case are now provided in Appendix E.1).  These analyses directly illustrate the bias towards sparsity in learned task-specific features (Fig. 1c, Appendix E fig. 5e for linear case, Fig. 2c, Appendix E fig. 6c+d for nonlinear case), and sparsity in features learned in the “nested feature selection” regime (Fig. 1h for linear case, Fig. 2h for nonlinear case) as a function of the weight rescaling factor.
>
> – We have integrated the analysis of appendix G, comparing the signatures of the nested feature selection regime between the teacher-student setting and the deep network experiments, into the main text, to reflect the greater emphasis on empirical results in the revised manuscript (Section 6, blue text, Fig. 3c).

---

### Official Review · Reviewer_gQGL · 2023-10-30

**Soundness:** 3 good
**Presentation:** 3 good
**Contribution:** 3 good
**Rating:** 8
**Confidence:** 4

**Summary:**

This paper provides a theoretical analysis characterizing the implicit regularization effects of multi-task learning (MTL) and pretraining + finetuning (PT+FT). The main contributions are:

- Derives analytical expressions for diagonal linear networks quantifying the implicit regularization biases of MTL (l1,2 norm) and PT+FT (Q norm) that encourage feature sharing and sparsity.
- Validates that MTL and PT+FT exhibit these regularizer-like behaviors for both linear and relu neural networks.
- Identifies a "nested feature learning" regime unique to PT+FT that enables sparse reuse of pretrained features. Proposes techniques to leverage this.
- Shows PT+FT is more flexible in "soft" feature sharing than MTL, leading to tradeoffs based on dataset size.
- Demonstrates applicability of theoretical insights on CIFAR-100 for improving auxiliary task learning.

Overall, this work provides novel theoretical analysis of the under-studied problem of characterizing implicit regularization in MTL and PT+FT. The paper opens promising new directions for better understanding and improvement of auxiliary task learning techniques widely used in deep learning.

**Strengths:**

- The paper focuses on an important open problem in deep learning theory. Using PT+FT is the de-facto go to tool for creating a modern DNN both in MLP and CV. Comparison to MTL with auxiliary task is a good way to analyze the available alternatives.
- Provides analytical characterization of implicit regularization in MTL and PT+FT for diagonal linear networks
- Validation of theory through experiments on synthetic and CIFAR100
- Identifies unique "nested feature learning" regime of PT+FT
- I think the experiments on the correlated features and the heterogeneous feature sharing are great, as this is probably the more relevant case for the real world applications.

**Weaknesses:**

- I would like to see more practical experiments that demonstrate the theoretical findings. In the given experiments the main task is 2 classes on CIFAR100 but it is not clear how the choice of the classes was made and weather it makes any difference which classes taken. I understand this is not the main focus of the paper but I think this section could be improved.
- The main focus of paper is on MSE loss, I think adding the CE loss focus could improve the paper.

**Questions:**

See weaknesses

---

> ### Author Response · Authors · 2023-11-16
> **Response to reviewer gQGL**
>
> We thank the reviewer for the positive evaluation of our work and suggestions for improvement.
>
> – We have updated the CIFAR experiments so that in each trial, the choice of the 2 classes used for the “main task” is randomized.  Moreover, we have scaled up the experiment to use all the samples available in the dataset.  We have also provided further corroboration (Appendix F) of the usefulness of the post-pretraining rescaling method we proposed – see the “response to all reviewers” for details.
>
> – We agree with the reviewer that an analysis of the cross-entropy loss case would be of interest and this is an active follow-up direction for us.  We provided some comments on the cross-entropy case in Appendix C of the original draft.  Unfortunately, we are unlikely to be able to provide a comprehensive analysis of the cross-entropy loss case during the discussion period – in particular, it requires a different theoretical analysis of PT+FT, which we leave to future work.

---

### Official Review · Reviewer_ToRY · 2023-10-30

**Soundness:** 3 good
**Presentation:** 3 good
**Contribution:** 3 good
**Rating:** 6
**Confidence:** 2

**Summary:**

The paper investigates the implicit regularization effects of multi-task learning (MTL) and pretraining followed by fine-tuning (PT+FT) in overparameterized neural networks. It makes a valuable contribution to our understanding of how auxiliary task learning influences neural network training. It offers practical insights for improving fine-tuning performance and provides a novel perspective on the dynamics of feature learning in overparameterized networks.

**Strengths:**

1. Novel Contribution: The paper introduces a novel concept of "nested feature learning," which is not captured by existing regimes ("lazy" or "rich") in the context of multi-task learning (MTL) and pre-training followed by fine-tuning (PT+FT). This new insight contributes to a deeper understanding of how neural networks operate in complex learning scenarios.

2. Inductive Biases in Different Scenarios: The authors explore the inductive biases in both linear and non-linear scenarios, providing a comprehensive analysis. This part looks good to me.

3. Real-world Experiment setting: The authors employ a combination of theoretical analysis and empirical experiments, including the use of real-world networks like ResNet-18 and challenging benchmarks like CIFAR-100. This ensures that the findings are grounded in practical relevance.

**Weaknesses:**

1. Sample Size and Repetition: The paper would benefit from addressing the potential limitation regarding sample size and repetition in the experiments. It is essential to clarify whether the results, as shown in Figure 3(f), are robust across different samples. The authors should consider repeating the experiments with various samples multiple times and reporting the mean performance along with confidence intervals. As it stands, it appears that the experiments were conducted only once, which may raise concerns about the reliability and generalizability of the findings. A more comprehensive analysis of the impact of different samples on the results would strengthen the paper's credibility and reliability.

2. Lack of Visualization and Evaluation Metrics: The paper could benefit from a more comprehensive presentation of results. It lacks visualization techniques to visually illustrate these biases. Visual representations can provide a clearer and more intuitive understanding of the observed phenomena.

**Questions:**

1. Regarding the experimental repetition, it's not clear why the author relies on results from a single experiment. Is this choice due to concerns about potential bias in the sampled data, or are there other reasons for not conducting multiple experiments to ensure robustness and reliability of the findings?

2. The paper mentions using "1024" samples for different testing scenarios. Could the authors provide more context on the rationale behind this sample size? What happens when the sample numbers are significantly larger? Are the "1024" samples allocated for each task, or is it 1024 samples shared among all tasks?

3. The figures corresponding to samples seem to lack clarity. The x-axis is labeled with "32 samples" and "256 samples," but it's not entirely clear which parts of the figures represent actual experimental results. Could the authors consider visually highlighting the data points with dots and connecting them with lines to provide a more explicit representation of the experimental results, making it easier for readers to interpret the findings?

---

> ### Author Response · Authors · 2023-11-16
> **Response to reviewer ToRY**
>
> We thank the reviewer for the positive evaluation of our work and suggestions for improvement.  We have attempted to address all the points raised in the review in our revised version of the manuscript:
>
> – **Error bars / number of repetitions run:** each experimental result shown is an average over many trials, using randomly sampled datasets and/or weight initializations, as appropriate. We meant to illustrate the trial-to-trial variability in our initial draft with error bars; however, this was not clear in our initial draft, as the error bars were small and the thickness of the lines obscured them.  We have revised the figures to make the confidence intervals more clear, and where they are not visible, it is because they are smaller than the thickness of the lines/points in the plots – we hope that the low variability addresses the reviewer’s concerns that the results may not be robust.  Moreover, the updated figures make it more clear which aspects of the plots are actual experimental results vs. interpolating lines, as suggested by the reviewer.
>
> – **Number of samples:** in our teacher-student, we use 1024 samples for the auxiliary task, and vary the number of main task samples (the number of main task samples is what is depicted on the x axes).  Ultimately the choice of 1024 is arbitrary and was made to allow a large number of experiments to be run efficiently while still being large enough to allow the network to achieve very good test-set performance on the auxiliary task.  We have run some experiments with more auxiliary task samples (and also more features in the ground-truth teacher network) which show that the key results of Fig. 3a are robust to scaling these parameters. We have added these results to Appendix D of the revised manuscript.
>
> – **Visualizations / evaluations:** we have provided several new analyses and figures in the revised manuscript that help illustrate the inductive biases we describe in more concrete terms.  Please see our “response to all reviewers” for a description of these analyses.

---

> > ### Comment · Reviewer_ToRY · 2023-11-22
> > **Acknowledgement**
> >
> > I acknowledge the contributions of this work and the author's rebuttal. I will maintain my score.

---

> > > ### Author Response · Authors · 2023-11-22
> > >
> > > We thank the reviewer for reading our response + revisions and maintaining their positive evaluation of the work.  As we believe we have addressed all the reviewer's original questions / listed weaknesses, we would love to know if there are any additional points we could clarify for the reviewer before the end of the discussion period that might cause them to increase their score.

---

### Author Response · Authors · 2023-11-16
**Response to all reviewers**

We thank the reviewers for their thoughtful comments on our manuscript.  We have updated our manuscript in response to reviewer’s comments and are also providing individual responses to reviewers.  Updates to the main text of the manuscript are highlighted in the revised manuscript in red text for reviewers’ convenience.  Here we highlight some supplementary analyses provided in the revised manuscript which may be of interest to all reviewers.

Two reviewers requested additional analyses / visualizations that more concretely illustrate the impact of the inductive biases we characterize in the paper. To that end, we have conducted several new analyses. In the revised manuscript, these analyses are given in appendices and referred to in the main text; if the paper is accepted, we plan to integrate the highlights of these results more gracefully into the main body of the paper in the camera-ready version.

– In **Appendix G**, we further analyze the signatures of the “nested feature selection” regime we describe in the paper.  Our goal was to find a signature of nested feature selection that is agnostic to the feature basis being considered, so that it can be measured in deep network representations, in order to assess whether the performance benefits associated with rescaling the network weights following pretraining really do arise because the network is pushed into the nested feature selection regime.  To do so, we characterize effects of finetuning in terms of its effect on the effective dimensionality of network representations, and the relationship between the representations pre/post finetuning (in particular, whether the pre- and post-finetuning representational dimensions are “shared” or distinct, according to a measure of “effective number of shared dimensions” between neural network representations introduced by Giaffar et al.).  We show that the student-teacher networks we characterized as being in the “nested feature selection” regime exhibit two key properties: (1) they learn lower-dimensional representations after finetuning than prior to finetuning, and (2) the post-finetuning representations inherit a subset of the representational dimensions of the pretrained network (i.e. the number of dimensions post-finetuning is not appreciably larger than the number of dimensions shared between the pre- and post-finetuning representations).  Strikingly, we find exactly the same phenomenology when we finetune our ResNet-18 model with rescaled weights (but not when we finetune the unscaled ResNet-18 model). This analysis provides evidence that the performance improvement of the rescaling technique really does arise from pushing the network into the nested feature selection regime.

– In **Appendix E**, we assess the extent to which learned representations in our shallow nonlinear network experiments concentrate on a sparse set of features by clustering the units of the network according to their normalized input weights. To briefly summarize the results, we show that networks we describe as having a sparsity-inducing inductive bias exhibit concentration of network units into a small number of clusters (thus effectively implementing a function that depends on a small set of features), while networks without a sparsity-inducing inductive bias do not.  We also show clusters align with the ground-truth teacher features, and that this alignment is improved by pretraining when main task and auxiliary task features overlap, clarifying a mechanistic underpinning of our finding that the inductive bias of PT+FT and MTL improves performance in such cases.

– In **Appendix F** we report additional experiments with deep convolutional networks / realistic tasks to corroborate our suggestion that rescaling network weights prior to finetuning can improve performance. Specifically, to further test the impact of the rescaling method, we have run new experiments using ImageNet as a pretraining dataset, varying both the network architecture (ResNet18 and VGG11) and the finetuning dataset (CIFAR-100 and STL10). In all these follow-up experiments, the results corroborate our initial findings, in that rescaling the pretrained network weights by a factor less than 1 improves finetuning performance.

---

### Author Response · Authors · 2023-11-23
**Additional revisions**

In response to one of the reviewers' reply to our original rebuttal, we have updated the text further to incorporate the following suggestions:

-- Clarifying the relationship between our theoretical statements and prior work in the abstract / introduction / related work section

-- Increase the formality of our presentation of the implicit regularization penalties associated with PT+FT and MTL, including a more precise statement of the assumptions required for these results to hold

-- Incorporate the direct analysis of feature sparsity in nonlinear networks from Appendix E into the main text (see updated fig. 2c,d,h) and include a similar analysis in the diagonal linear case (see updated fig. 1c,d,h)

-- Incorporate the highlights of the additional analysis of the nested feature selection regime in Appendix G into the main text (see updated fig. 3c).

---

### Meta-Review · Area_Chair_BaNY · 2023-12-10

**Metareview:**

This paper explores the inductive biases from learning auxiliary tasks, either through multi-task learning (MTL) or pretraining and finetuning (PT+FT). The study, focusing on two-layer diagonal linear networks, finds that both MTL and PT+FT encourage feature sharing between tasks and favor sparsity in task-specific features. The authors show that during FT networks operate as a mix of lazy and feature learning regime. The authors suggest that their work reveals that PT+FT tends to learn features partly similar to those needed for the auxiliary task, while MTL uses almost identical features for both tasks. Consequently, MTL is more effective with limited data for the primary task, whereas PT+FT excels with more data. They also carryout experiments in deep architectures trained on image classification tasks to support this.

The reviewers thought that this research direction is interesting and provides insights into the impact of auxiliary task learning and suggests strategies for its more effective use, and that the authors have nice experimental results. They did raise various concerns including Sample size and repetition, lack of visualization and evaluation metrics, novelty and significance, and presentation of the paper. The authors response alleviated some of these concerns. In particular the authors agreed that their theoretical contributions are limited but posited that the main contributions of the work is empirical. This however did not satisfy the negative reviewer who pointed out that the paper does actually make claims about theoretical contributions (the authors alleviated that in a second round) and that the standard of empirical validations should be higher if the focus is mostly on empirical observations. The latter issue IMO is valid and has not been fully resolved. I think more thorough experimental analysis (comparisons, ablation studies, etc) is needed to tease out the effect of different components and for this paper to be suitable for publication. That said, I personally enjoyed the paper and would happily advocate for in in a future ML venue after a thorough revision.

**Justification For Why Not Higher Score:**

I think more thorough experimental studies are needed

**Justification For Why Not Lower Score:**

N/A

---

### Decision · Program_Chairs · 2024-01-16

Reject